# TRACED: Transition-aware Regret Approximation with Co-learnability for Environment Design

**Geonwoo Cho    Jaegyun Im    Jihwan Lee    Hojun Yi    Sejin Kim**[*] **Sundong Kim**[*]
Gwangju Institute of Science and Technology
{gwcho.public, jaegyun.public, qhddl2650, hojun172635, sjkim7822, sdkim0211}@gmail.com

## Abstract

Generalizing deep reinforcement learning agents to unseen environments remains a significant challenge. One promising solution is Unsupervised Environment Design (UED), a co-evolutionary framework in which a teacher adaptively generates tasks with high learning potential, while a student learns a robust policy from this evolving curriculum. Existing UED methods typically measure learning potential via regret, the gap between optimal and current performance, approximated solely by value-function loss. Building on these approaches, we introduce the transition-prediction error as an additional term in our regret approximation. To capture how training on one task affects performance on others, we further propose a lightweight metric called Co-Learnability. By combining these two measures, we present Transition-aware Regret Approximation with Co-learnability for Environment Design (TRACED). Empirical evaluations show that TRACED produces curricula that improve zero-shot generalization over strong baselines across multiple benchmarks. Ablation studies confirm that the transition-prediction error drives rapid complexity ramp-up and that Co-Learnability delivers additional gains when paired with the transition-prediction error. These results demonstrate how refined regret approximation and explicit modeling of task relationships can be leveraged for sample-efficient curriculum design in UED.
https://geonwoo.me/traced/

## 1 Introduction

Deep reinforcement learning (RL) has achieved remarkable success in games, continuous control, and robotics (Sutton et al., 1998). Ideally, we want agents that generalize robustly to a broad range of unseen environments. However, hand-crafting a training distribution that captures all real-world variability is intractable, and agents often overfit even large training sets, performing poorly out-of-distribution (Kirk et al., 2023; Korkmaz, 2024).

Unsupervised Environment Design (UED) tackles this by adapting the curriculum: a teacher module generates training tasks that challenge the student agent (Dennis et al., 2020). A popular class of UED methods measures task difficulty by regret, the difference between the optimal return and the agent's achieved return, and uses this metric to guide curriculum design (Dennis et al., 2020; Jiang et al., 2021b;a; Parker-Holder et al., 2022; Azad et al., 2023; Mediratta et al., 2023; Erlebach & Cook, 2024). Unfortunately, genuine regret requires knowing each environment's optimal $Q^*$, which is infeasible in complex domains. Existing approaches, therefore, resort to coarse proxies such as Positive Value Loss (PVL) or maximum observed return (MaxMC) (Rutherford et al., 2024). In this paper, we refine regret estimation by augmenting PVL with a transition-prediction error term that captures how poorly a learned model predicts the environment's dynamics. This combined signal provides a more faithful approximation of regret and a sharper basis for curriculum design.

We further introduce a metric called *Co-Learnability* to quantify how training on one task benefits others. For instance, consider three 100-word vocabulary tasks, Spanish, English, and Japanese, whose transfer patterns differ: because Spanish and English share many cognates, learning Spanish

---

[*]Corresponding authors.

accelerates lexical access and boosts English accuracy (Ramírez et al., 2013; Costa et al., 2000; Lemhöfer & Dijkstra, 2004), reflecting high Co-Learnability; in contrast, Japanese is typologically distant (Chiswick & Miller, 2005), so gains from Japanese may not transfer to English, reflecting low Co-Learnability. We present a lightweight estimator of Co-Learnability that leverages observed changes in approximated regret, avoiding any additional modeling overhead in the UED loop.

We propose **TRACED** (Transition-aware Regret Approximation with Co-Learnability for Environment Design), which combines refined regret and Co-Learnability to yield a task-priority landscape (Figure 1). We evaluate TRACED on procedurally generated MiniGrid (MG) (Chevalier-Boisvert et al., 2023) and Bipedal-Walker (BW) (Romac et al., 2021). In MG, we compare against Domain Randomization (DR) (Jakobi, 1997), PLR$^{\perp}$ (Jiang et al., 2021b), ADD (Chung et al., 2024), and ACCEL (Parker-Holder et al., 2022) (the strongest baseline). In BW, we additionally include the state-of-the-art method CENIE (Teoh et al., 2024).

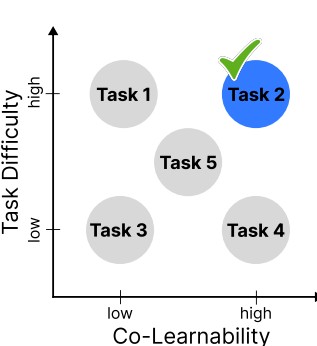

Figure 1: **Task Priority Landscape.** Task with high difficulty and high Co-Learnability are scheduled with the highest priority in the curriculum.

TRACED surpasses all baselines in mean solved rate across 12 MiniGrid mazes, BipedalWalker, and even the extreme Perfect-Maze variants. Ablation studies show that the transition-prediction error term accelerates curriculum ramp-up, while Co-Learnability provides additional gains when combined with our regret term. An analysis of curriculum evolution indicates that TRACED progressively increases task difficulty from easy to challenging. Taken together, these results chart a path toward more sample-efficient UED by coupling refined regret approximation with explicit modeling of task relationships.

We provide the full implementation of TRACED to facilitate the reproduction of our main results at https://github.com/Cho-Geonwoo/TRACED.

## 2 PRELIMINARIES

### 2.1 UNDERSPECIFIED PARTIALLY OBSERVABLE MDPS (UPOMDPS)

We model our environments as *underspecified partially observable Markov decision processes* (UP-OMDPs) following Dennis et al. (2020). A UPOMDP is a tuple $\mathcal{M} = \langle A, O, \Theta, S, P_0, P_T, \mathcal{I}, \mathcal{R}, \gamma \rangle$, where $A$ is a finite set of actions, $S$ is a latent state space, and $O$ is an observation space. The observation function $\mathcal{I} : S \to \Delta(O)$ generates each observation $o_t$ given the true state $s_t$, the reward function $\mathcal{R} : S \times A \to \mathbb{R}$ and discount factor $\gamma \in [0, 1)$ are shared across all levels. Crucially, $\Theta$ is a set of *underspecified parameters* that distinguish individual "levels": for each $\theta \in \Theta$, the initial state is drawn from $P_0(\theta) \in \Delta(S)$ and transitions follow $P_T(s_{t+1} \mid s_t, a_t, \theta)$.

At each time step $t$, the agent observes $o_t \sim \mathcal{I}(s_t)$ and selects an action $a_t$ according to a trajectory-conditioned policy $\pi(a_t|o_0, a_0, \ldots, o_t)$. For a fixed level $\theta$, the utility of policy $\pi$ is the expected discounted return $U_\theta(\pi) = \mathbb{E}\left[\sum_{t=0}^{T} \gamma^t r_t\right], r_t = \mathcal{R}(s_t, a_t)$ with the expectation taken over both the stochastic dynamics and the policy's choices. We denote an optimal policy on level $\theta$ by $\pi_\theta^\star \in \arg\max_\pi U_\theta(\pi)$.

### 2.2 UNSUPERVISED ENVIRONMENT DESIGN (UED)

UED provides a series of levels with unknown parameters that are used to produce task environments automatically as a curriculum for the agent so as to efficiently train a single generalist policy $\pi_\phi$ across the entire parameter space $\Theta$ (Dennis et al., 2020). Recent UED methods maximize regret of the agent to generate a distribution of environments that guide effective learning (Jiang et al., 2021b; Parker-Holder et al., 2022; Teoh et al., 2024). The regret of policy $\pi$ is defined as the difference in expected reward between optimal policy $\pi^*$ and policy $\pi$, $Regret_\theta(\pi) = U_\theta(\pi^*) - U_\theta(\pi)$.

Because the true optimal policy $\pi^\star$ is unknown in complex environments, existing methods approximate the instantaneous regret using proxy metrics. For example, PLR$^{\perp}$ (Jiang et al., 2021a)

evaluates two such proxies: *Positive Value Loss* (PVL) and *Maximum Monte Carlo* (MaxMC). PVL estimates the instantaneous regret as the average positive part of the Generalized Advantage Estimation (GAE)-based Temporal Difference (TD) errors over an episode. PVL for an episode $\tau$ of length $T$ is defined as

$$\text{PVL}(\tau) = \frac{1}{T} \sum_{t=0}^{T} \max\Big(\sum_{k=t}^{T} (\gamma\lambda)^{k-t} \delta_k, 0\Big) \tag{1}$$

where $\gamma$ is the discount factor, $\lambda$ is the GAE coefficient and $\delta_t = r_t + \gamma V(s_{t+1}) - V(s_t)$ is the one-step TD-error at timestep $t$. MaxMC uses the highest undiscounted return observed on the task instead of a bootstrap target. Other criteria such as policy entropy, one-step TD error, GAE, policy min-margin, and policy least-confidence have also been evaluated in curriculum learning contexts (Jiang et al., 2021b).

## 3 TRACED: TRANSITION-AWARE REGRET APPROXIMATION WITH CO-LEARNABILITY FOR ENVIRONMENT DESIGN

TRACED improves UED via (i) a regret approximation combining value and transition-prediction errors and (ii) a Co-Learnability measure. These yield a unified *Task Priority* score that governs new-task generation and replay sampling. We first present the motivating regret decomposition, then the curriculum derived from the approximated regret.

### 3.1 REGRET APPROXIMATION VIA TRANSITION PREDICTION LOSS

Since regret quantifies the difficulty an agent experiences on a task, improving its approximation can yield more accurate difficulty estimates and, in turn, more effective curricula. We approximate the one-step regret at a state-action pair $(s, a)$ via the following decomposition:

$$
\begin{aligned}
\text{Regret}(s, a) &= V^*(s) - Q^\pi(s, a) \\
&= V^*(s) - \hat{V}^*(s) + \hat{V}^*(s) - Q^\pi(s, a) \\
&= \underbrace{V^*(s) - \hat{V}^*(s)}_{\text{(i) Value estimation error}} + \underbrace{r(s, a^*) - r(s, a)}_{\text{(ii) Reward gap}} \\
&\quad + \gamma \underbrace{\left( \mathbb{E}_{s'' \sim \hat{P}(\cdot|s, a^*)}\big[\hat{V}^*(s'')\big] - \mathbb{E}_{s' \sim P(\cdot|s, a)}\big[V^\pi(s')\big] \right)}_{\text{(iii) Future value gap}},
\end{aligned}
\tag{2}
$$

*Notation.* A hat (e.g., $\hat{V}^*(s)$) indicates an empirical/learned estimator of the optimal value; $Q^*, V^*$ are the optimal value functions; $Q^\pi, V^\pi$ are the value functions under policy $\pi$; $P$ is the true transition kernel and $\hat{P}$ is a learned transition model; $\gamma$ is the discount factor.

Eq. 2 makes clear that Positive Value Loss (PVL), which evaluates only the accuracy of the empirical value estimator and thus corresponds to term (i), is insufficient as a proxy for regret. The future-value gap in term (iii) is influenced not only by value-function error but also by mismatch between the learned dynamics $\hat{P}$ and the true dynamics $P$. Motivated by this, we augment PVL with a transition-prediction error term, explicitly accounting for model-environment dynamics mismatch when approximating regret.

**Definition 1 (Average Transition Prediction Loss).** Train a transition dynamics estimator $f_\phi$, implemented as a recurrent model, to minimize a one-step reconstruction loss $L_{\text{trans}}(s_t, a_t)$ between the observed next state $s_{t+1}$ and the prediction $\hat{s}_{t+1} = f_\phi(s_t, a_t)$. Detailed design choices for the transition model are provided in Appendix I.3. Over an episode $\tau = (s_0, a_0, \ldots, s_T)$, define

$$\text{ATPL}(\tau) = \frac{1}{T} \sum_{t=0}^{T} L_{\text{trans}}(s_t, a_t). \tag{3}$$

Combine the two estimates into a single scalar:

$$\widehat{\text{Regret}}(\tau) = \text{PVL}(\tau) + \alpha \, \text{ATPL}(\tau), \tag{4}$$

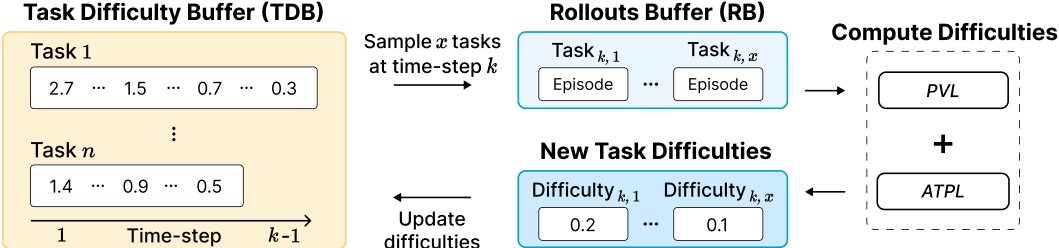

Figure 2: **Task Difficulty Calculation Workflow.** The Task Difficulty Buffer (TDB) records each task's history of approximated regret. The agent interacts with sampled tasks to collect episode trajectories, which are stored in the Rollouts Buffer (RB). For each trajectory, we compute the Positive Value Loss (PVL) and the Average Transition-Prediction Loss (ATPL). Their sum produces the updated task difficulty (approximated regret), which is appended to the TDB to refresh each sampled task's stored difficulty.

where $\alpha > 0$ balances value versus transition terms.

By explicitly incorporating both components, our regret approximation more faithfully captures task difficulty and, in turn, yields more effective curricula. For each sampled task instance $\tau$, we compute $\widehat{\text{Regret}}(\tau)$ and append it to the task difficulty buffer (TDB), which is subsequently used to drive curriculum updates (Figure 2). Detailed theoretical analysis of the relationship between term (iii) in Eq. 2 and ATPL is provided in Appendix A, where we show that the dynamics-induced component of the future-value gap is upper-bounded by ATPL. This establishes ATPL as a principled correction term for regret approximation.

### 3.2 TASK PRIORITY CONSTRUCTION

A central challenge in UED is to decide which tasks to present to the agent at each step (Hughes et al., 2024). Intuitively, we want to (1) focus on tasks that remain challenging, while (2) exploiting tasks whose training yields transfer to others. To this end, we introduce two complementary quantities, *Task Difficulty* and *Co-Learnability*, and combine them into a single *Task Priority* score. In the following sections, time $t$ indexes the teacher's curriculum-selection cycles (not environment time).

**Definition 2 (Task Difficulty).** Let $s_i(k) = \max\{ s \leq k : \text{TDB}(i, s) \text{ exists} \}$ denote the most recent time at or before $k$ when task $i$ was sampled, so that its approximated regret was stored at time $s_i(k)$. We define the task difficulty of task $i$ at time $k$ as

$$\text{TaskDifficulty}(i, k) = \begin{cases} \text{TDB}\big(i,\ s_i(k)\big), & \text{if } s_i(k) \text{ is finite,} \\ 0, & \text{if } i \text{ has never been sampled before } t. \end{cases} \tag{5}$$

This ensures that $\text{TaskDifficulty}(i, k)$ always reflects the most recent approximated regret for task $i$, with larger values indicating greater remaining challenge. Note that a difficulty value for task $i$ is well-defined at time $k$ only if the task has been sampled at least once on or before $k$. This difficulty estimate becomes available for use when constructing the curriculum at time $k+1$.

**Definition 3 (Co-Learnability).** Beyond difficulty, we wish to capture how training on one task accelerates progress on others. Let $\mathcal{T}_{k+1}$ be the set of tasks replayed at time $k + 1$. We define the Co-Learnability (CL) of task i at time k as

$$\text{CoLearnability}_i(k) = \frac{1}{|\mathcal{T}_{k+1}|} \sum_{j \in \mathcal{T}_{k+1}} \big[ \text{TaskDifficulty}(j, k) - \text{TaskDifficulty}(j, k + 1) \big] \tag{6}$$

which measures the average reduction in the difficulty of replayed tasks when task $i$ is selected at time $k$. In principle, the true marginal contribution could be computed via Shapley values; however, due to computational constraints we approximate task $i$'s effect on reducing other tasks' difficulty using this surrogate. A positive Co-Learnability value indicates that visiting $i$ yields transfer benefits. Since $\text{CoLearnability}_i(k)$ depends on difficulty updates that occur at time $k + 1$, it becomes available beginning at timestep $k + 2$.

Figure 3: **Method Workflow Overview.** The three panels in the figure depict: (1) *Task Sampling*: levels are drawn from the buffer based on their priority scores. (2) *Priority Update*: we recompute each level's priority based on our task priority definition (Eq. 7). (3) *Task Mutation*: the lowest-priority levels are mutated into new variants and reinserted into the buffer.

**Definition 4 (Task Priority).** We combine difficulty and Co-Learnability into a scalar score and then apply a *rank* transform, which replaces raw values with their relative order (e.g., the largest value receives rank 1, the next largest rank 2, etc.), as in ACCEL (Parker-Holder et al., 2022):

$$\text{TaskPriority}(i, t) = \text{Rank}\Big(\text{TaskDifficulty}(i, t) + \beta \, \text{CoLearnability}(i, t)\Big) \tag{7}$$

where $\beta > 0$ trades off challenge versus transfer. At each step, we sample tasks inversely to their priority (e.g., $p(i \mid t) \propto 1/\text{TaskPriority}(i, t)$), so lower ranks (higher priority) are selected more often.

The rank transform mitigates the influence of outliers by discarding the absolute magnitude of raw scores and retaining only their relative ordering. Without this transformation, a single task with an anomalously large difficulty or Co-Learnability value can dominate the sampling distribution, collapsing the curriculum toward that task.

### 3.3 OVERALL UED WORKFLOW

The overall UED algorithm follows the ACCEL loop (Parker-Holder et al., 2022), with the sole change that task scoring uses our Task Priority (Eq. 7) in place of Positive Value Loss (PVL) alone (Figure 3). At each curriculum update time $t$, with probability $d$ we sample a new level uniformly at random, and with probability $1 - d$ we sample a level from the replay buffer according to its Task Priority score. Early in training, when the buffer is empty, we sample levels uniformly at random for a warm-up period. The warm-up length matches ACCEL's setting.

After training the agent on the selected level, we approximate its regret using the agent's value-related loss and the transition-prediction loss (for Task Difficulty), and append the result to the task difficulty buffer. Co-Learnability is updated following Eq. 6 after the next curriculum step, once the newly visited levels have had their difficulties updated. The updated Task Difficulty and Co-Learnability define Task Priority via Eq. 7, which then governs both new-task sampling and prioritized replay at future times. The algorithm alternates between sampling new tasks and prioritized replay until policy performance converges. The full procedure is given in Algorithm 1.

## 4 EXPERIMENTS

We evaluate TRACED on two procedurally generated domains: MiniGrid (MG) and BipedalWalker (BW). In both environments, we compare against DR, PLR$^\perp$, ACCEL (our primary baseline, since TRACED builds directly upon it), and ADD. In BW only, we additionally include the recent method CENIE (Teoh et al., 2024). While SFL (Rutherford et al., 2024) is not included in the main experimental suite, we provide a comprehensive comparison against SFL in the multi-agent JaxNav environment in Appendix R, following its original evaluation protocol. Detailed descriptions of all baselines can be found in Appendix L.

For each domain, we track the emergent complexity of the sampled curricula during training and evaluate zero-shot test performance on held-out levels. To summarize test results, we report the median, interquartile mean (IQM), mean, and optimality gap using the rliable library (Agarwal et al., 2021). The optimality gap measures how far an algorithm falls short of a target performance level,

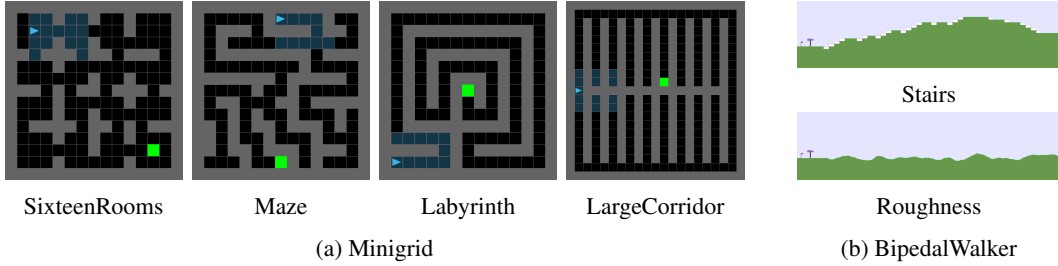

| SixteenRooms | Maze | Labyrinth | LargeCorridor | Roughness |

(a) Minigrid            (b) BipedalWalker

Figure 4: **Held-out Evaluation Environments.** (a) Example held-out MiniGrid mazes for zero-shot evaluation: 4 tasks are shown (see Appendix I.1 for all 12 task definitions). (b) Example held-out BipedalWalker terrains for zero-shot evaluation: 2 tasks are shown (see Appendix I.2 for all 6 task definitions).

beyond which further gains are deemed negligible. Accordingly, higher IQM and lower optimality gap values indicate better performance. Shaded regions in all plots denote 95% confidence intervals.

All methods use Proximal Policy Optimization (PPO) (Schulman et al., 2017) as the student agent. Following prior work (Parker-Holder et al., 2022; Teoh et al., 2024), we plot performance versus the number of PPO updates and report the corresponding environment interactions per update in Appendix G. Our only deviation from ACCEL is the number of PPO workers (i.e., parallel trajectory collectors), chosen for computational constraints: MiniGrid uses 16 workers instead of 32, and BipedalWalker uses 4 instead of 16. Consequently, some baseline scores may differ from those in the ACCEL paper. However, Appendix B.4 shows that TRACED has no worker-specific tuning and that reducing workers does not render the comparison unfair. We report TRACED at 10k updates (rather than 20k) because single-seed, long-horizon runs (45k PPO updates) on MiniGrid reveal post-convergence oscillations in both TRACED and ACCEL, and reporting earlier avoids this confound (Appendix C). Additional hyperparameters and architectures are provided in Appendix I.

## 4.1 PARTIALLY OBSERVABLE NAVIGATION

We evaluate our curriculum design on a partially observable maze navigation domain based on MiniGrid (Chevalier-Boisvert et al., 2023), as in prior UED work. Four representatives are shown in Figure 4. The agent observes a 147-dimensional pixel observation and is trained for up to 20k PPO updates.

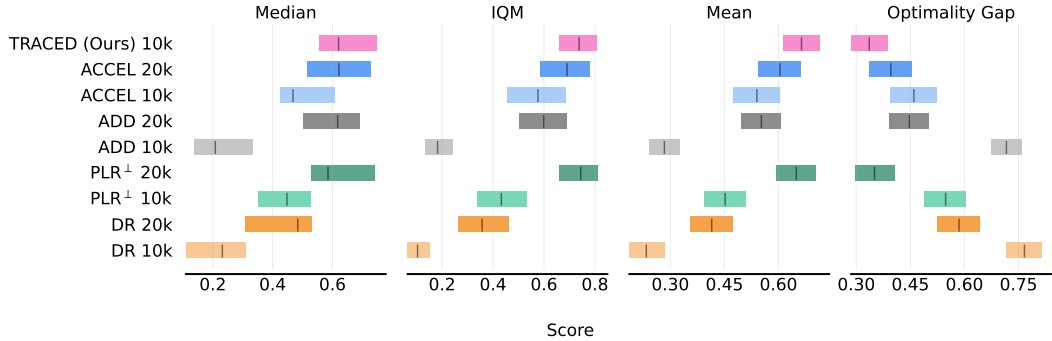

Figure 5: **Zero-Shot Transfer Performance in MiniGrid.** Aggregated solved rates on held-out MiniGrid mazes after 10k and 20k PPO updates. TRACED at 10k updates outperforms baselines at 20k updates.

Our method achieves a superior solved rate at only 10k updates, matching or exceeding the 20k update performance of all baselines (Figure 5). In particular, TRACED's median solved rate at 10k surpasses ACCEL at 20k, and its IQM leads the field, indicating that the majority of runs benefit rapidly from our combined regret and Co-Learnability scoring.

Compared to ACCEL, TRACED halves the wall-clock training time while maintaining equivalent or better transfer performance (Table 1). Even relative to ACCEL at the same 10k update budget,

Table 1: **Wall-clock training time comparison.** Average training duration (hours $\pm$ SE over 10 runs) on the MiniGrid domain.

| TRACED 10k | ACCEL 10k | ACCEL 20k | ADD 10k | ADD 20k | PLR$^\perp$ 10k | PLR$^\perp$ 20k | DR 10k | DR 20k |
|---|---|---|---|---|---|---|---|---|
| $13.78 \pm 0.36$ | $12.94 \pm 0.66$ | $26.58 \pm 0.76$ | $22.48 \pm 0.27$ | $45.08 \pm 0.29$ | $14.87 \pm 0.62$ | $31.83 \pm 1.36$ | $5.82 \pm 0.12$ | $12.41 \pm 0.18$ |

TRACED incurs a 6% computational overhead yet delivers a 22% relative increase in mean solved rate. Taken together, these metrics demonstrate that TRACED not only accelerates learning but also delivers more consistent and reliable zero-shot transfer across diverse MiniGrid mazes. Detailed per-task zero-shot results are provided in Appendix Q, and the number of environment interactions per PPO update for each method is in Appendix G.

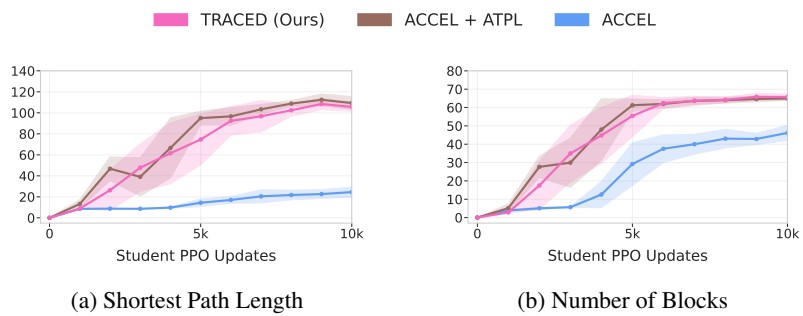

(a) Shortest Path Length        (b) Number of Blocks

Figure 6: **Emergent maze complexity metrics.** Shortest path length and number of blocks both grow faster under TRACED (pink) than ACCEL (blue). This faster ramp-up indicates that our curriculum more effectively escalates difficulty in lockstep with agent learning.

To analyze emergent environment complexity, we track two structural metrics for each generated maze: (i) the length of the shortest solution path and (ii) the number of obstacles (Figure 6). Curves are averaged over 10 seeds with shaded 95% confidence intervals. Both metrics increase substantially faster under TRACED than under ACCEL, indicating that our priority scoring more effectively separates easy from challenging tasks and yields a steadily escalating curriculum. Notably, ACCEL+ATPL also drives complexity upward far faster than ACCEL alone and closely tracks TRACED, demonstrating that the transition-prediction component on its own contributes strongly to complexity ramp-up.

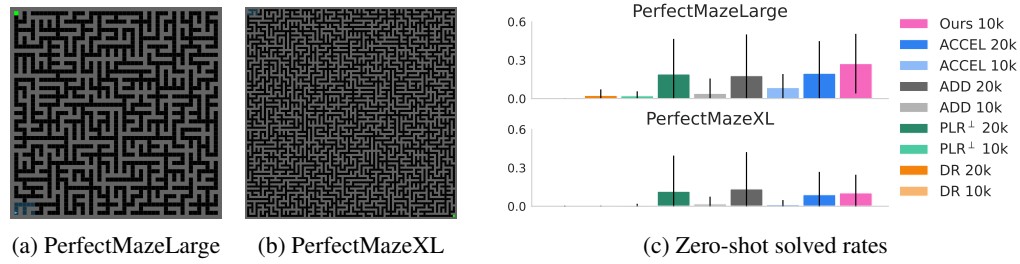

(a) PerfectMazeLarge   (b) PerfectMazeXL      (c) Zero-shot solved rates

Figure 7: **PerfectMaze Evaluation. (a), (b)** Two held-out maze instances, PerfectMazeLarge and PerfectMazeXL, used for zero-shot testing. **(c)** Zero-shot solved rates. TRACED achieves the highest 10k performance on PerfectMazeLarge and closely matches the best 20k performance on PerfectMazeXL.

To stress-test our curriculum on extremely large, procedurally generated mazes, we introduce two PerfectMaze benchmarks. PerfectMazeLarge consists of $51 \times 51$ grids with a maximum episode length exceeding 5k steps, while PerfectMazeXL scales this to $100 \times 100$ grids. Figure 7 shows representative levels from each variant. We evaluate zero-shot transfer performance, measuring the mean success rate over 100 episodes per seed (10 seeds total).

On PerfectMazeLarge, TRACED achieves the highest 10k solved rate ($27\% \pm 23\%$), outperforming ACCEL's best 20k rate ($20\% \pm 25\%$) and far exceeding ADD, DR and PLR$^\perp$ (Figure 7). On the even more complex PerfectMazeXL, ACCEL narrowly leads at $12\% \pm 28\%$ after 20k updates, with TRACED close behind at $10\% \pm 14\%$ after just 10k updates, demonstrating that TRACED scales to extremely large mazes. Complete per-baseline numerical results for both benchmarks are reported in Appendix Q, and the full set of aggregate metrics (Median, IQM, Mean, and Optimality Gap) for PerfectMaze is shown in Figure 21.

## 4.2 WALKING IN CHALLENGING TERRAIN

We further validate our curriculum in the continuous-control BipedalWalkerHardcore environment from OpenAI Gym (Brockman et al., 2016), as modified by Wang et al. (2019). This domain features a procedurally generated terrain controlled by eight parameters, terrain roughness, pit gap frequency, stump height, stair spacing, etc., that jointly determine locomotion difficulty. We consider the complete set of eight parameters in our design space. Figure 4 illustrates two representative terrains with varying stair heights and surface roughness. We evaluate zero-shot transfer on six held-out test terrains over 100 episodes each, averaged over five random seeds.

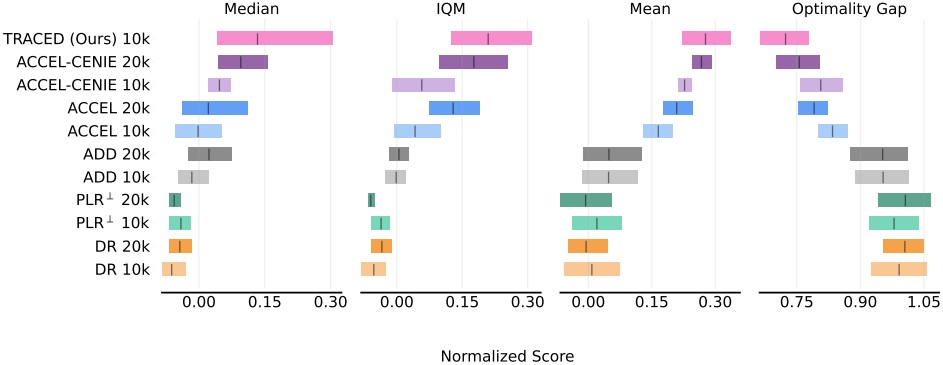

Figure 8: **Aggregate zero-shot performance on BipedalWalker terrains.** All scores are normalized by the maximum return of 300 in the BipedalWalker domain. TRACED at 10k updates (pink) matches or exceeds ACCEL-CENIE at 20k updates (purple) across all metrics.

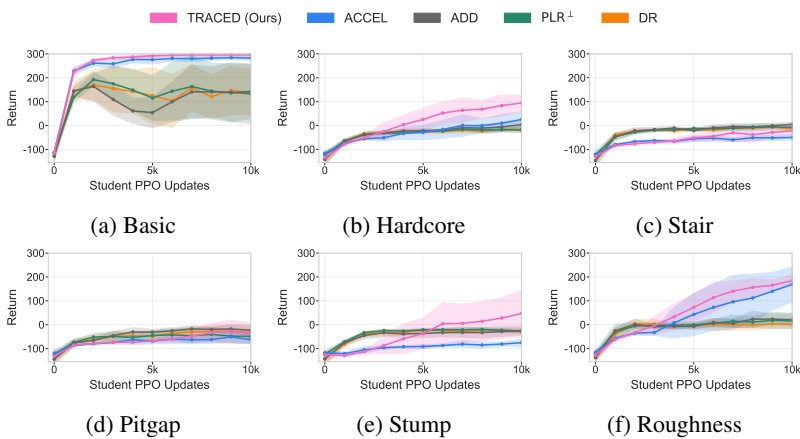

Figure 9: **Return progression on BipedalWalker terrains.** TRACED consistently outperforms baselines.

TRACED delivers consistent gains across all four aggregate metrics, median, interquartile mean (IQM), mean, and optimality gap, on the six held-out BipedalWalker terrains (Figure 8). After only 10k updates, it already outperforms all baselines evaluated at 20k. As in MiniGrid, TRACED

also reaches its peak performance in roughly half the wall-clock time of ACCEL on BipedalWalker (Appendix G). Moreover, TRACED consistently surpasses ACCEL in zero-shot returns across all terrains throughout training (Figure 9), further underscoring the effectiveness of its curriculum design. Detailed numerical results are provided in Appendix Q.

## 4.3 ABLATION STUDY

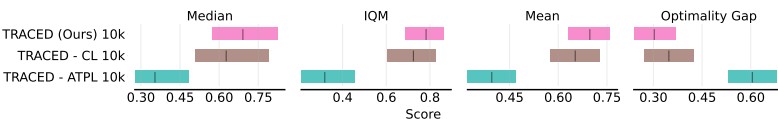

Figure 10: **Ablation Study on TRACED.** Both ATPL and CL are important design choices.

To isolate component contributions, we compare TRACED at 10k updates with two ablations (Figure 10): *ATPL only* (TRACED − CL) and *Co-Learnability only* (TRACED − ATPL), each evaluated with five seeds. Results are reported following Agarwal et al. (2021). On MiniGrid, TRACED outperforms both variants across four metrics, indicating that each component contributes meaningfully to performance. The same trend holds on BipedalWalker (Appendix B). Ablations on the scaling factors $\alpha$ and $\beta$ further show that both elements play significant roles in TRACED and that carefully balancing ATPL and CL can yield additional gains (Appendix B.1). Detailed numerical results are provided in Appendix Q.

## 4.4 ANALYSIS ON CURRICULUM PROGRESSION

Table 2: **Proportions (%) of Easy, Moderate, and Challenging levels in the level buffer during PPO updates.**

| Method | Difficulty | 0k | 5k | 10k | 15k | 20k |
|---|---|---|---|---|---|---|
| **ACCEL** | **Easy** | 100 | 100 | 79.2 | 26.7 | 9.2 |
| | **Moderate** | 0 | 0 | 20.8 | 73.3 | 90.8 |
| | **Challenging** | 0 | 0 | 0 | 0 | 0 |
| **TRACED** | **Easy** | 100 | 72 | 24.5 | 19.5 | 11.2 |
| | **Moderate** | 0 | 25.7 | 60.9 | 68 | 77.3 |
| | **Challenging** | 0 | 2.3 | 14.6 | 12.5 | 11.5 |

To analyze how the curriculum evolves, we examine the BipedalWalker level buffer over time and categorize each generated level into three difficulty bands: Easy, Moderate, and Challenging, based on environment hyperparameters. Following Teoh et al. (2024), we set thresholds for Stump Height (2.4), Pit Gap (6), Ground Roughness (4.5), and Stairs Height (5). Levels exceeding no thresholds are labeled Easy, those exceeding exactly one threshold are Moderate, and those exceeding at least two thresholds are Challenging. Table 2 reports the proportion of each difficulty in the buffer at different PPO update steps.

TRACED progressively shifts mass from Easy to Moderate and Challenging, introducing nontrivial proportions of Challenging levels by 10k (14.6%) and maintaining them thereafter (11.5% at 20k). In contrast, ACCEL never surfaces Challenging levels even at 20k (0% throughout), with the buffer dominated by Moderate levels by 20k (90.8%). This steady escalation under TRACED aligns with its faster convergence and stronger performance already at 10k updates. The respective contributions of ATPL and Co-Learnability (CL) to curriculum progression are analyzed in Appendix D, and a visualization of level evolution is provided in Appendix E.

## 5 CONCLUSION

In this paper, we introduced a Unsupervised Environment Design (UED) method with two key components: (i) an explicit transition-prediction error term for regret approximation, and (ii) a lightweight Co-Learnability metric that captures cross-task transfer effects. By integrating these into

the standard generator-replay loop, TRACED produces curricula that escalate environment complexity in tandem with agent learning.

Empirically, we demonstrated on two procedurally generated domains (MiniGrid navigation and BipedalWalker) that TRACED outperforms baselines (DR, PLR$^{\perp}$, ADD, ACCEL), including CE-NIE, using only half the training updates. We showed superior zero-shot transfer success rates, faster growth in structural complexity, and scalability to extremely large mazes. Ablation studies confirmed that each component is essential: ATPL drives the primary complexity ramp-up, while Co-Learnability yields gains when paired with our regret estimates.

Looking forward, Co-Learnability offers a simple, computationally light mechanism for capturing inter-task influences and could be further refined via more sophisticated causal estimators or learned models. More broadly, any RL setting that relies on regret-approximation stands to benefit from incorporating transition-prediction error, providing an easy-to-implement boost in sample efficiency. We anticipate that these ideas will inspire future work on adaptive curricula, advanced editing mechanisms, and broader applications of regret-guided exploration in open-ended learning environments.

ACKNOWLEDGMENTS

This work was supported by IITP (RS-2024-00445087; 10%, RS-2025-25410841; 10%, No. 2019-0-01842; 10%), NRF (RS-2024-00451162; 20%, RS-2024-00454000; 20%, RS-2025-25419920; 20%), GIST (KH0870; 10%) funded by the Ministry of Science and ICT, Korea. Computing resource were supported by GIST SCENT-AI, AICA, KISTI, and NIPA.

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

## A  THEORETICAL ANALYSIS OF ATPL

We restate the one-step regret decomposition (Eq. 2 in the main paper):

$$\text{Regret}(s,a) = \underbrace{V^*(s) - \hat{V}^*(s)}_{\text{(i) Value estimation error}} + \underbrace{r(s,a^*) - r(s,a)}_{\text{(ii) Reward gap}}$$

$$+ \gamma \underbrace{\left( \mathbb{E}_{s'' \sim \hat{P}(\cdot|s,a^*)} [\hat{V}^*(s'')] - \mathbb{E}_{s' \sim P(\cdot|s,a)} [V^\pi(s')] \right)}_{\text{(iii) Future value gap}},$$

Our objective is to isolate and control the portion of term (iii) that arises specifically from the mismatch between the true transition kernel $P$ and the learned transition model $\hat{P}$. In particular, we focus on the discrepancy between the value predictions under $P$ and $\hat{P}$, and show that this dynamics-induced error can be bounded by the transition loss used in ATPL.

### A.1  ASSUMPTIONS

**Assumption 1 (Lipschitz value function).** Let $(\mathcal{S}, d)$ be the state space equipped with a metric $d(\cdot, \cdot)$. The approximate optimal value function $\widehat{V}^* : \mathcal{S} \to \mathbb{R}$ is $L_V$-Lipschitz with respect to $d$, i.e.,

$$\left| \widehat{V}^*(s_1) - \widehat{V}^*(s_2) \right| \leq L_V \, d(s_1, s_2), \qquad \forall s_1, s_2 \in \mathcal{S}.$$

**Assumption 2 (Loss dominating the metric).** Let $\ell : \mathcal{S} \times \mathcal{S} \to [0, \infty)$ be the reconstruction loss used to train the transition model (e.g., $\ell(x, y) = \|x - y\|^2$). We assume that there exists a constant $C_\ell > 0$ such that

$$d(x, y) \leq C_\ell \, \ell(x, y), \qquad \forall x, y \in \mathcal{S}.$$

**Definition 5 (Coupling-based transition loss).**

For each state-action pair $(s, a)$, let $P(\cdot \mid s, a)$ and $\hat{P}(\cdot \mid s, a)$ denote the true and learned transition kernels, respectively. A *coupling* of $P(\cdot \mid s, a)$ and $\hat{P}(\cdot \mid s, a)$ is any joint distribution $\Gamma_{s,a}$ on $\mathcal{S} \times \mathcal{S}$ such that its marginals satisfy

$$(s^+, \hat{s}^+) \sim \Gamma_{s,a} \quad \Longrightarrow \quad s^+ \sim P(\cdot \mid s, a), \qquad \hat{s}^+ \sim \hat{P}(\cdot \mid s, a).$$

Given such a coupling $\Gamma_{s,a}$, we define the one-step transition loss as

$$L_{\text{trans}}(s, a) := \mathbb{E}_{(s^+, \hat{s}^+) \sim \Gamma_{s,a}} \left[ \ell(s^+, \hat{s}^+) \right].$$

### A.2  BOUNDING THE DYNAMICS MISMATCH TERM

**Lemma 1.** *Define the dynamics mismatch term*

$$\Delta_{\text{dyn}}(s, a) := \left| \mathbb{E}_{\hat{P}(\cdot|s,a)} [\widehat{V}^*(s'')] - \mathbb{E}_{P(\cdot|s,a)} [\widehat{V}^*(s')] \right|.$$

*Under Assumptions 1 and 2, for any coupling $\Gamma_{s,a}$ of $P(\cdot \mid s, a)$ and $\hat{P}(\cdot \mid s, a)$, we have*

$$\Delta_{\text{dyn}}(s, a) \leq L_V \, C_\ell \, L_{\text{trans}}(s, a).$$

*Proof.* Let $(s^+, \hat{s}^+) \sim \Gamma_{s,a}$ be a coupling of $P(\cdot \mid s, a)$ and $\hat{P}(\cdot \mid s, a)$, i.e., $s^+ \sim P(\cdot \mid s, a)$ and $\hat{s}^+ \sim \hat{P}(\cdot \mid s, a)$. Then, by the definition of the marginal distributions,

$$\mathbb{E}_{\hat{P}(\cdot|s,a)} [\widehat{V}^*(s'')] = \mathbb{E}_{(s^+, \hat{s}^+) \sim \Gamma_{s,a}} [\widehat{V}^*(\hat{s}^+)],$$

and

$$\mathbb{E}_{P(\cdot|s,a)} [\widehat{V}^*(s')] = \mathbb{E}_{(s^+, \hat{s}^+) \sim \Gamma_{s,a}} [\widehat{V}^*(s^+)].$$

Hence

$$
\begin{aligned}
\Delta_{\mathrm{dyn}}(s,a) &= \left| \mathbb{E}_{(s^+,\hat{s}^+)\sim\Gamma_{s,a}}\big[\widehat{V}^*(\hat{s}^+)\big] - \mathbb{E}_{(s^+,\hat{s}^+)\sim\Gamma_{s,a}}\big[\widehat{V}^*(s^+)\big] \right| \\
&= \left| \mathbb{E}_{(s^+,\hat{s}^+)\sim\Gamma_{s,a}}\big[\widehat{V}^*(\hat{s}^+) - \widehat{V}^*(s^+)\big] \right| \\
&\le \mathbb{E}_{(s^+,\hat{s}^+)\sim\Gamma_{s,a}}\Big[\big|\widehat{V}^*(\hat{s}^+) - \widehat{V}^*(s^+)\big|\Big] \\
&\le L_V\,\mathbb{E}_{(s^+,\hat{s}^+)\sim\Gamma_{s,a}}\big[d(s^+,\hat{s}^+)\big] \qquad \text{(by Lipschitzness of } \widehat{V}^*) \\
&\le L_V\,C_\ell\,\mathbb{E}_{(s^+,\hat{s}^+)\sim\Gamma_{s,a}}\big[\ell(s^+,\hat{s}^+)\big] \qquad \text{(by the dominance } d \le C_\ell\ell) \\
&= L_V\,C_\ell\,L_{\mathrm{trans}}(s,a).
\end{aligned}
$$

$\square$

## A.3  REGRET APPROXIMATION

We compare two regret proxies:

$$
\widehat{\mathrm{Regret}}_{\mathrm{PVL}}(\tau) := \mathrm{PVL}(\tau), \qquad \widehat{\mathrm{Regret}}_{\mathrm{ATPL}}(\tau) := \mathrm{PVL}(\tau) + \alpha\,\mathrm{ATPL}(\tau).
$$

**Theorem 1** (Regret Approximation Improvement with ATPL). *For any trajectory $\tau$,*

$$
\left| \mathrm{Regret}(\tau) - \widehat{\mathrm{Regret}}_{\mathrm{PVL}}(\tau) \right| \le C_0 + L_V C_\ell\,\mathrm{ATPL}(\tau),
$$

*where $C_0$ collects the remaining terms in the regret decomposition that do not depend on the transition mismatch. Furthermore,*

$$
\left| \mathrm{Regret}(\tau) - \widehat{\mathrm{Regret}}_{\mathrm{ATPL}}(\tau) \right| \le C_0 + \left| L_V C_\ell - \alpha \right|\,\mathrm{ATPL}(\tau).
$$

*In particular, choosing $\alpha = L_V C_\ell$ yields a strictly tighter worst-case approximation bound than PVL alone.*

*Proof.* Lemma 1 bounds the dynamics-induced component of the regret by $L_V C_\ell\,\mathrm{ATPL}(\tau)$. Substituting this bound into the regret decomposition and subtracting $\alpha\,\mathrm{ATPL}(\tau)$ gives the inequality. $\square$

**Interpretation.**  PVL captures only the value-estimation component of regret and assigns zero weight to the error arising from transition-model mismatch. Lemma 1 shows that this missing component is controlled by ATPL. Theorem 1 then establishes that augmenting PVL with $\alpha\,\mathrm{ATPL}$ reduces the worst-case approximation whenever $\alpha$ is chosen to match the scale of the transition mismatch (i.e., $L_V C_\ell$). Thus, ATPL provides a principled correction for the future-value error, yielding a more faithful regret proxy for unsupervised environment design.

# B  ADDITIONAL EXPERIMENTS

## B.1  ABLATION STUDY ON THE SCALING FACTOR

In this ablation, we keep every setting in Appendix P fixed except for one of the weight-scaling factors, $\alpha$ (ATPL weight), $\beta$ (Co-Learnability weight). Results are averaged over five random seeds and reported as mean $\pm$ standard error on 12 held-out MiniGrid tasks. Each method is evaluated after 10k PPO updates.

Table 3: **Ablation study on the scaling factor.** Comparing different fixed projection Weight baselines ($\alpha = 0.0$, $\alpha = 100.0$, $\beta = 0.0$, $\beta = 100.0$) and TRACED ($\alpha = 1.0$, $\beta = 1.0$). **Bold** indicates the best; underline indicates the second-best.

| Environment | $\alpha$=0.0 | $\alpha$=100.0 | $\beta$=0.0 | $\beta$=100.0 | TRACED |
|---|---|---|---|---|---|
| 16Rooms | **0.81** $\pm$ 0.1 | 0.06 $\pm$ 0.04 | 0.73 $\pm$ 0.09 | 0.74 $\pm$ 0.17 | 0.79 $\pm$ 0.19 |
| 16Rooms2 | **0.94** $\pm$ 0.05 | 0.0 $\pm$ 0.0 | 0.28 $\pm$ 0.2 | 0.59 $\pm$ 0.15 | 0.72 $\pm$ 0.17 |
| SimpleCrossing | 0.84 $\pm$ 0.04 | 0.43 $\pm$ 0.08 | 0.86 $\pm$ 0.04 | 0.82 $\pm$ 0.04 | **0.89** $\pm$ 0.01 |
| FourRooms | 0.45 $\pm$ 0.05 | 0.23 $\pm$ 0.04 | 0.41 $\pm$ 0.05 | **0.52** $\pm$ 0.03 | 0.47 $\pm$ 0.02 |
| SmallCorridor | 0.42 $\pm$ 0.24 | 0.12 $\pm$ 0.06 | **0.62** $\pm$ 0.21 | 0.51 $\pm$ 0.12 | 0.49 $\pm$ 0.17 |
| LargeCorridor | 0.44 $\pm$ 0.26 | 0.04 $\pm$ 0.02 | 0.54 $\pm$ 0.16 | **0.56** $\pm$ 0.18 | 0.5 $\pm$ 0.14 |
| Labyrinth | 0.5 $\pm$ 0.29 | 0.0 $\pm$ 0.0 | 0.49 $\pm$ 0.25 | 0.19 $\pm$ 0.18 | **1.0** $\pm$ 0.0 |
| Labyrinth2 | 0.61 $\pm$ 0.2 | 0.0 $\pm$ 0.0 | 0.2 $\pm$ 0.17 | 0.2 $\pm$ 0.2 | **0.98** $\pm$ 0.01 |
| Maze | **0.74** $\pm$ 0.21 | 0.0 $\pm$ 0.0 | 0.18 $\pm$ 0.07 | 0.39 $\pm$ 0.24 | 0.59 $\pm$ 0.17 |
| Maze2 | **0.56** $\pm$ 0.26 | 0.0 $\pm$ 0.0 | 0.51 $\pm$ 0.21 | 0.14 $\pm$ 0.12 | 0.42 $\pm$ 0.19 |
| Maze3 | 0.51 $\pm$ 0.27 | 0.0 $\pm$ 0.0 | 0.59 $\pm$ 0.22 | 0.51 $\pm$ 0.21 | **0.86** $\pm$ 0.07 |
| PerfectMaze(M) | 0.4 $\pm$ 0.16 | 0.01 $\pm$ 0.02 | 0.4 $\pm$ 0.06 | 0.34 $\pm$ 0.07 | **0.66** $\pm$ 0.11 |
| **Mean** | 0.6 $\pm$ 0.1 | 0.07 $\pm$ 0.01 | 0.5 $\pm$ 0.09 | 0.46 $\pm$ 0.05 | **0.7** $\pm$ 0.04 |

Deviating from defaults on either $\alpha$ or $\beta$ reduces the solved rates in most domains (Table 3). For both ignoring one of the weighted terms ($\alpha = 0.0$ or $\beta = 0.0$) and using excessively large weight ($\alpha = 100.0$ or $\beta = 100.0$), solved rates show slight improvement for some individual tasks, but statistical drops for overall performance. The default TRACED weights show the best performance compared to fixed projection baselines. This result shows that an effective balance between ATPL and Co-Learnability is needed.

## B.2  ABLATION STUDY ON ATPL AND CO-LEARNABILITY

In this ablation, we evaluate TRACED by removing each of its two components in turn. Table 4 reports zero-shot solve rates on six held-out BipedalWalker tasks after 10k PPO updates. Results are averaged over five random seeds and reported as mean $\pm$ standard error.

Table 4: **Ablation study on ATPL and Co-Learnability.** Comparing ACCEL (10k), TRACED - ATPL (10k), TRACED - CL (10k), and TRACED (10k, $\alpha = 1.5$, $\beta = 0.6$). **Bold** indicates the best result per task; underline indicates the second-best.

| Environment | ACCEL 10k | TRACED - ATPL 10k | TRACED - CL 10k | TRACED 10k |
|---|---|---|---|---|
| Basic | 281.65 $\pm$ 5.25 | 282.61 $\pm$ 4.64 | 286.75 $\pm$ 5.01 | **293.67** $\pm$ 3.56 |
| Hardcore | 37.59 $\pm$ 15.0 | 53.77 $\pm$ 20.8 | 48.35 $\pm$ 23.74 | **86.83** $\pm$ 17.96 |
| Stairs | -38.71 $\pm$ 10.54 | -43.99 $\pm$ 12.99 | -34.1 $\pm$ 8.44 | **-29.0** $\pm$ 10.4 |
| PitGap | -65.07 $\pm$ 7.57 | -72.92 $\pm$ 13.95 | **-5.2** $\pm$ 66.89 | -39.26 $\pm$ 11.42 |
| Stump | -79.18 $\pm$ 5.45 | -38.06 $\pm$ 39.88 | -19.89 $\pm$ 64.11 | **34.16** $\pm$ 54.58 |
| Roughness | 161.72 $\pm$ 28.36 | 191.75 $\pm$ 25.07 | 191.68 $\pm$ 4.99 | **193.29** $\pm$ 21.6 |
| **Mean** | 49.67 $\pm$ 11.24 | 62.19 $\pm$ 11.21 | 77.93 $\pm$ 17.58 | **89.95** $\pm$ 12.95 |

TRACED achieves the top score in five of six tasks (Basic, Hardcore, Stairs, Stump, Roughness) and ranks second in the remaining one. The ATPL only variant (TRACED - CL) leads to the first place in one task (PitGap) and second place in two tasks (Basic, Stump), while the Co-Learnability only variant (TRACED - ATPL) only reaches the second place in two tasks (Hardcore, Roughness). Removing both components (the ACCEL baseline) ranks second on one (Stairs). These results confirm that both transition-prediction error and Co-Learnability are essential to TRACED's high and stable performance across diverse tasks.

### B.3 HYPERPARAMETER SENSITIVITY ANALYSIS

We analyze impact of varying the ATPL weight $\alpha$ and the Co-Learnability weight $\beta$ in TRACED across the 12 held-out MiniGrid environments and six held-out BipedalWalker terrains. Results are reported for a single random seed due to computational constraints.

Table 5: **Effect of hyperparameter weights** $(\alpha, \beta)$ **on MiniGrid environments.** Evaluations are performed under varying $(\alpha, \beta)$ settings. **Bold** indicates the best; underline indicates the second-best.

| Environment | (0.75, 1.0) | (1.25, 1.0) | (1.0, 0.75) | (1.0, 1.25) | (1.25, 0.75) | **TRACED** (1.0, 1.0) |
|---|---|---|---|---|---|---|
| 16Rooms | 0.26 | 0.90 | **1.00** | 0.96 | **1.00** | 0.79 ± 0.19 |
| 16Rooms2 | 0.00 | 0.05 | 0.08 | **0.78** | 0.10 | 0.72 ± 0.17 |
| SimpleCrossing | 0.82 | 0.84 | 0.85 | 0.71 | 0.78 | **0.89** ± 0.01 |
| FourRooms | **0.68** | 0.36 | 0.52 | 0.39 | 0.51 | 0.47 ± 0.02 |
| SmallCorridor | **0.55** | 0.00 | 0.01 | 0.00 | 0.01 | 0.49 ± 0.17 |
| LargeCorridor | **0.94** | 0.02 | 0.01 | 0.00 | 0.01 | 0.50 ± 0.14 |
| Labyrinth | 0.98 | **1.00** | **1.00** | **1.00** | **1.00** | **1.00** ± 0.00 |
| Labyrinth2 | **1.00** | 0.73 | 0.76 | **1.00** | 0.95 | 0.98 ± 0.01 |
| Maze | 0.76 | 0.16 | 0.39 | 0.00 | **1.00** | 0.59 ± 0.17 |
| Maze2 | 0.07 | 0.00 | 0.94 | **1.00** | 0.98 | 0.42 ± 0.19 |
| Maze3 | 0.92 | **1.00** | 0.10 | 0.89 | 0.93 | 0.86 ± 0.07 |
| PerfectMaze(M) | 0.56 | 0.38 | 0.55 | 0.55 | **0.85** | 0.66 ± 0.11 |
| **Mean** | 0.63 | 0.45 | 0.52 | 0.61 | 0.67 | **0.70** ± 0.04 |

In MiniGrid environments, four of the six $(\alpha, \beta)$ settings outperform the 10k-update baselines (Appendix Q), and even the weakest setting matches the performance of PLR$^{\perp}$ at 10k (Table 5). These results indicate that TRACED provides consistently strong zero-shot generalization across a broad range of hyperparameters, without extensive tuning.

Table 6: **Effect of hyperparameter Weights** $(\alpha, \beta)$ **on BipedalWalker environments.** Evaluations are performed under varying $(\alpha, \beta)$ settings. **Bold** indicates the best; underline indicates the second-best.

| Environment | (1.25, 0.6) | (1.75, 0.6) | (1.5, 0.45) | (1.5, 0.75) | **TRACED** (1.5, 0.6) |
|---|---|---|---|---|---|
| Basic | 282.06 | 291.04 | 281.32 | 279.37 | **293.67**±3.56 |
| Hardcore | -7.60 | **115.87** | 29.55 | 50.10 | 86.83±17.96 |
| Stairs | -52.78 | **-21.94** | -42.37 | -66.02 | -29.0±10.4 |
| PitGap | **12.96** | -52.72 | -52.21 | -122.54 | -39.26±11.42 |
| Stump | -100.95 | -40.11 | -74.83 | -73.93 | **34.16**±54.58 |
| Roughness | 148.22 | 187.43 | **244.57** | 185.83 | 193.29±21.6 |
| **Mean** | 46.99 | 79.93 | 64.34 | 42.13 | **89.95**±12.95 |

In BipedalWalker, three of the configurations outperform all 10k-update baselines (Appendix Q) (Table 6). Even the mean return across all five $(\alpha, \beta)$ choices ($\approx 64.9$) exceeds every baseline. These results demonstrate that TRACED delivers consistently strong zero-shot generalization across a wide range of weight settings, without requiring extensive hyperparameter tuning.

### B.4 ABLATION STUDY ON THE NUMBER OF WORKERS

We analyze the effect of the number of PPO workers, which determines how many agents collect trajectories in parallel. The only difference between our experimental setup and ACCEL (Parker-Holder et al., 2022) is the worker count: we use 16 workers for MiniGrid and 4 for BipedalWalker, whereas the original ACCEL paper uses 32 and 16, respectively. To ensure this configuration does not unfairly disadvantage TRACED, we additionally evaluate MiniGrid with 32 workers. Even under this setting, TRACED consistently outperforms ACCEL, indicating that our gains are robust to the PPO worker configuration (Table 7).

Table 7: **Ablation study on Number of Workers. Bold** indicates the best; underline indicates the second-best.

| Environment | ACCEL 5k | ACCEL 10k | TRACED 5k | TRACED 10k |
|---|---|---|---|---|
| 16Rooms | $0.37 \pm 0.17$ | $0.78 \pm 0.14$ | $\mathbf{0.95} \pm 0.02$ | $\underline{0.93} \pm 0.07$ |
| 16Rooms2 | $0.20 \pm 0.20$ | $\underline{0.77} \pm 0.23$ | $0.29 \pm 0.14$ | $\mathbf{0.91} \pm 0.04$ |
| SimpleCrossing | $0.65 \pm 0.09$ | $0.73 \pm 0.07$ | $\underline{0.78} \pm 0.02$ | $\mathbf{0.88} \pm 0.05$ |
| FourRooms | $0.33 \pm 0.08$ | $\underline{0.45} \pm 0.04$ | $0.33 \pm 0.04$ | $\mathbf{0.59} \pm 0.01$ |
| SmallCorridor | $\underline{0.52} \pm 0.21$ | $0.27 \pm 0.17$ | $\mathbf{0.62} \pm 0.31$ | $0.10 \pm 0.04$ |
| LargeCorridor | $\underline{0.42} \pm 0.17$ | $0.36 \pm 0.30$ | $\mathbf{0.49} \pm 0.24$ | $0.16 \pm 0.13$ |
| Labyrinth | $0.15 \pm 0.15$ | $0.49 \pm 0.29$ | $\underline{0.67} \pm 0.33$ | $\mathbf{0.91} \pm 0.09$ |
| Labyrinth2 | $0.0 \pm 0.0$ | $0.51 \pm 0.29$ | $\underline{0.65} \pm 0.20$ | $\mathbf{1.0} \pm 0.0$ |
| Maze | $0.33 \pm 0.33$ | $0.36 \pm 0.32$ | $\underline{0.95} \pm 0.04$ | $\mathbf{0.99} \pm 0.01$ |
| Maze2 | $0.01 \pm 0.01$ | $\mathbf{0.74} \pm 0.23$ | $0.22 \pm 0.19$ | $\underline{0.60} \pm 0.20$ |
| Maze3 | $0.14 \pm 0.13$ | $0.48 \pm 0.27$ | $\underline{0.63} \pm 0.29$ | $\mathbf{1.0} \pm 0.0$ |
| PerfectMaze(M) | $0.20 \pm 0.09$ | $0.48 \pm 0.07$ | $\underline{0.58} \pm 0.10$ | $\mathbf{0.63} \pm 0.13$ |
| **Mean** | $0.28 \pm 0.10$ | $0.54 \pm 0.09$ | $\underline{0.60} \pm 0.06$ | $\mathbf{0.72} \pm 0.09$ |

# C    LONG-TERM ANALYSIS ON TRACED

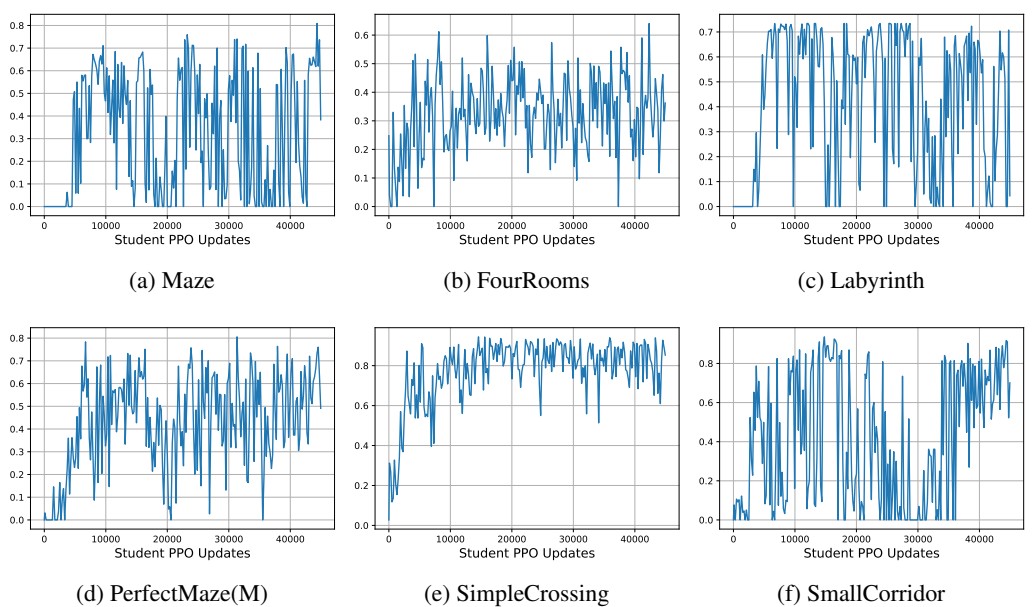

Figure 11: **TRACED Solved-Rate Time Series.** Solved rate progression on MiniGrid tasks plotted over 0-45k PPO updates.

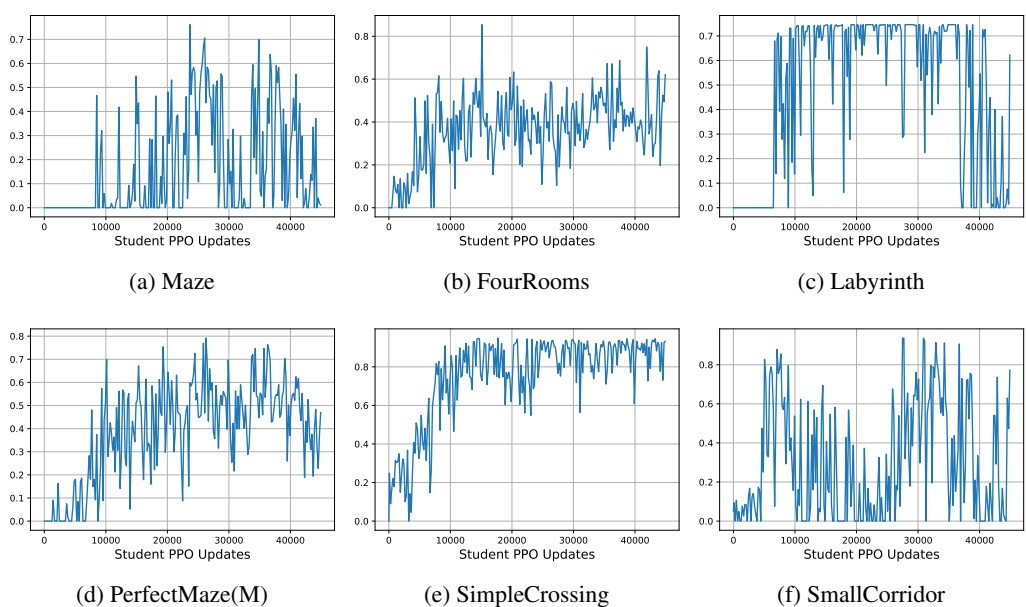

Figure 12: **ACCEL Solved-Rate Time Series.** Solved rate progression on MiniGrid tasks plotted over 0-45k PPO updates.

To study long-horizon behavior, we ran single-seed, 45k PPO updates for both TRACED and AC-CEL in the MiniGrid environment. Both methods exhibit post-convergence oscillations in success rate (i.e., solved rate). TRACED reaches near-peak success rapidly, but its performance subsequently oscillates rather than remaining perfectly stable (Figure 11). The same post-peak fluctuations appear under the ACCEL curriculum (Figure 12): although ACCEL attains its maximum more

slowly, it shows a similar up-and-down pattern thereafter. These oscillations likely reflect inherent instability of the RL policy on held-out levels, and may be further influenced by ACCEL's mutation dynamics, rather than a pathology introduced by our task-prioritization strategy. For this reason, we report TRACED's performance at 10k updates (rather than 20k), before long-horizon oscillations confound comparisons. Mitigating these fluctuations in the student policy is an important direction for future work.

# D    VISUALIZING CURRICULUM DYNAMICS

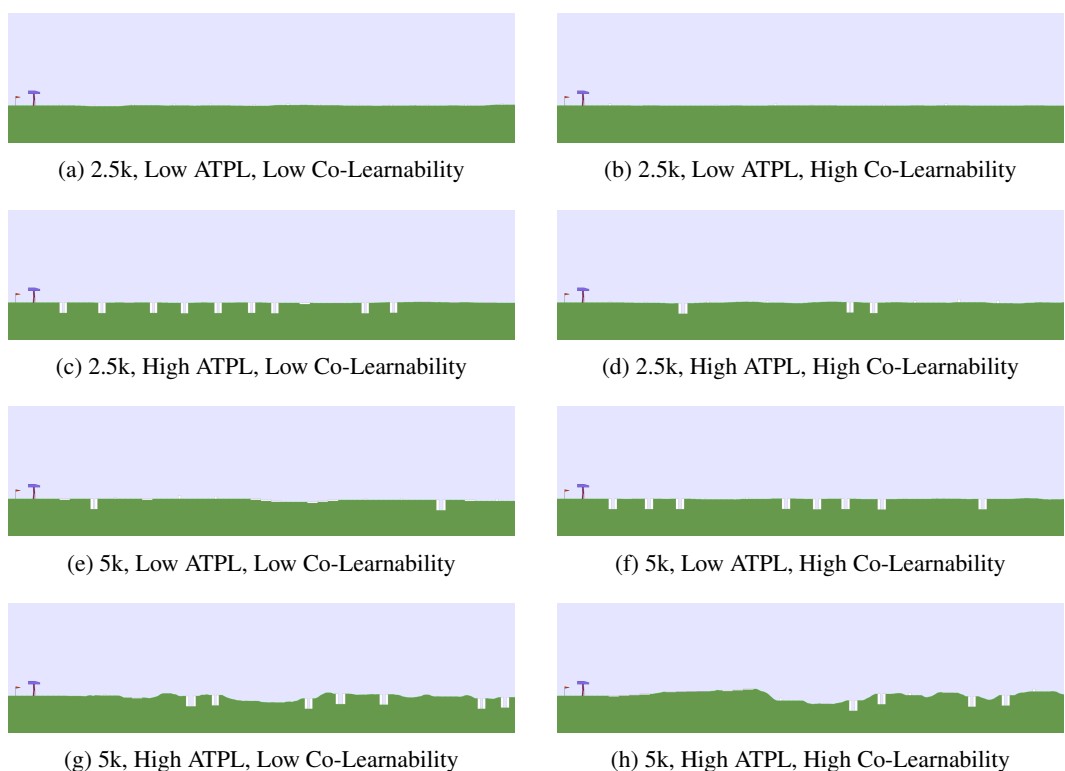

(a) 2.5k, Low ATPL, Low Co-Learnability

(b) 2.5k, Low ATPL, High Co-Learnability

(c) 2.5k, High ATPL, Low Co-Learnability

(d) 2.5k, High ATPL, High Co-Learnability

(e) 5k, Low ATPL, Low Co-Learnability

(f) 5k, Low ATPL, High Co-Learnability

(g) 5k, High ATPL, Low Co-Learnability

(h) 5k, High ATPL, High Co-Learnability

Figure 13: **Representative terrains from BipedalWalker selected by ATPL and co-learnability at two training stages.** The top two rows show terrains after 2.5k PPO updates, and the bottom two rows after 5k updates.

Figure 13 illustrates the joint impact of ATPL and Co-Learnability on terrain selection at 2.5k and 5k PPO updates. ATPL alone captures the raw challenge level: low-ATPL terrains (first and third rows) remain relatively smooth, while high-ATPL terrains (second and fourth rows) feature larger gaps and steeper bumps. As training proceeds from 2.5k to 5k updates, the overall terrains become systematically harder, demonstrating TRACED's ability to ramp up difficulty in lockstep with the agent's improving skills. Co-Learnability then refines this progression by filtering out extremes: low Co-Learnability (left column) tends to generate either trivial or overwhelmingly difficult levels that can stall learning, whereas high Co-Learnability (right column) favors intermediate-difficulty terrains that transfer more effectively across tasks. Together, these visualizations confirm that ATPL drives a steadily increasing complexity trajectory, while Co-Learnability smooths it to sustain robust, transferable learning throughout the curriculum.

# E    LEVEL EVOLUTION

## E.1    VISUALIZATION OF LEVEL EVOLUTION IN MINIGRID

The visualization of how the Minigrid environment evolves as the number of blocks increases (Figure 14). Each step of the evolutionary process produces an edited level that has a high learning efficiency.

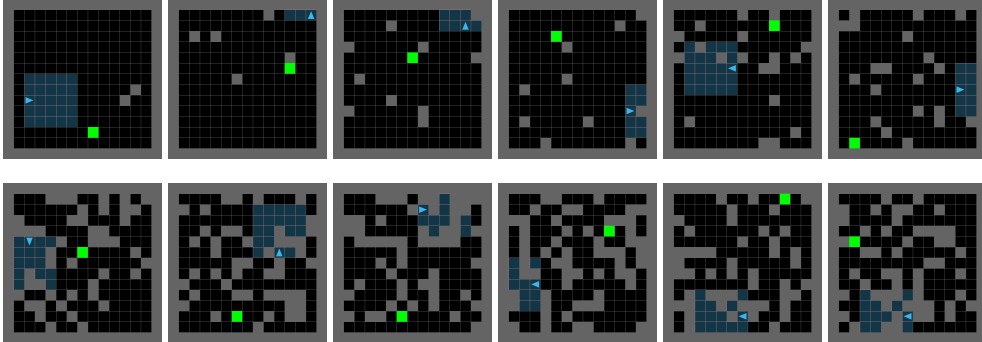

Figure 14: **Visualization of a single level's evolving progression in the MiniGrid environment.** Starting from top-left, ending bottom-right. This progress is automatically designed by our TRACED algorithm.

## E.2    EVOLUTION OF LEVELS IN BIPEDALWALKER

Figure 15 shows the complexity metric results trained by three methods, TRACED (Ours), ACCEL + ATPL, ACCEL. Starting with plain terrain (near zero point), all three methods guide levels with more complex terrain. With respect to (a) Stump height, (b) Stump height high, (e) Stair height step metrics, the results show TRACED and ACCEL + ATPL quickly evolve levels compared to ACCEL. This rapid level increase leads to shorter wall-clock time and higher performance across various test environments.

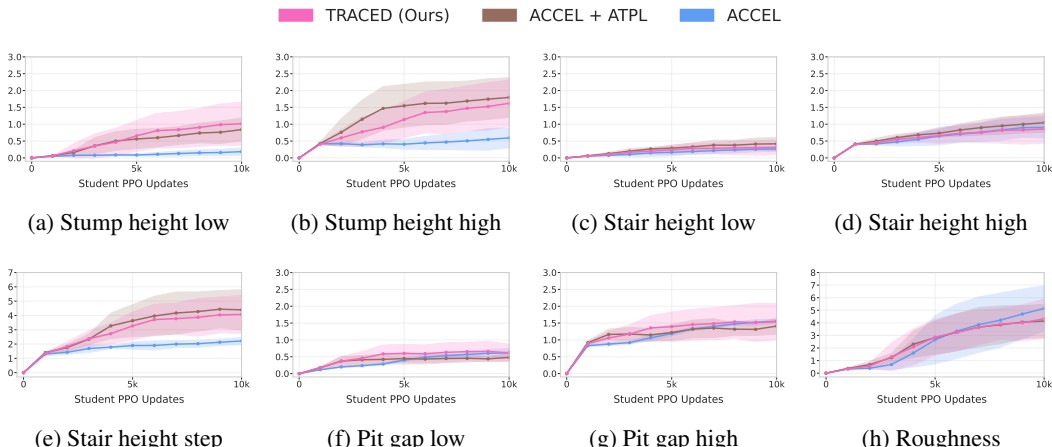

Figure 15: **Emergent BipedalWalker terrain complexity metrics.** Aspect of (a) Stump height low, (b) Stump height high, and (e) Stair height step, complexity grows faster under TRACED (Ours, pink) or ACCEL + ATPL (brown) than under ACCEL (blue). The result is averaged over five random seeds. This faster ramp-up indicates that our curriculum more effectively escalates difficulty in lockstep with agent learning.

### E.3 Visualization of Level Evolution in BipedalWalker

Figure 16 shows the visualization of how the environment evolves as the complexity of the task increases. The process starts from a plain terrain and gradually evolves to a level of terrain with increasingly complex parameters.

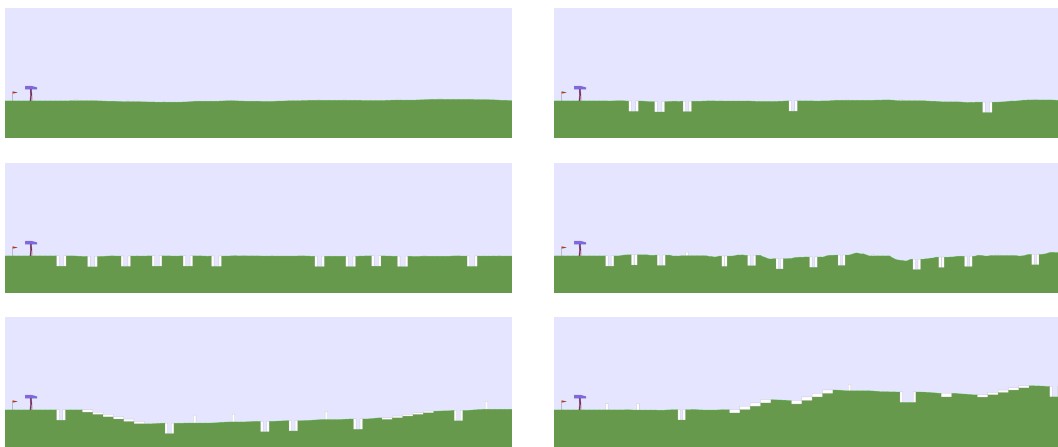

Figure 16: **Visualization of the level evolving progression in the BipedalWalker environment.** Starting from top-left, ending bottom-right. In this example, starting with plain terrain, the pits are created and their number increases, then the roughness increases, then stairs and stumps are created and their number, width, and height increase. This progress is automatically designed by our TRACED algorithm.

## F Agent Trajectory Visualizations Across Environments

### F.1 MiniGrid

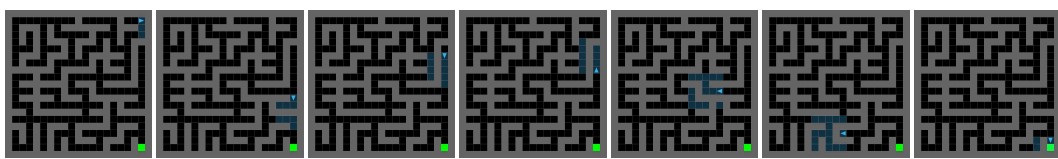

Figure 17: **Agent trajectory visualization in the PerfectMaze evaluation task.** TRACED enables efficient path planning and robust generalization to challenging maze environments, as demonstrated by the agent's successful navigation to the goal (green).

### F.2 BipedalWalker

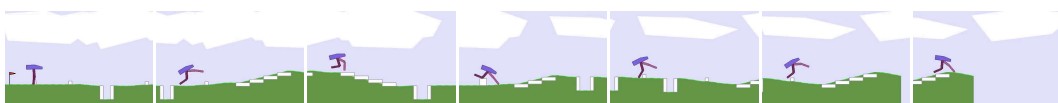

Figure 18: **Agent trajectory visualization in the BipedalWalkerHardcore.** TRACED agent successfully overcomes various combined obstacles, including stairs, pit gaps, stumps, and roughness, demonstrating robust generalization to complex environments.

# G    EFFICIENCY ANALYSIS

## G.1    WALL-CLOCK TRAINING TIME IN BIPEDALWALKER

Table 8 reports wall-clock training times on the BipedalWalker domain, in hours (mean $\pm$ s.e.) over five runs at 10k and 20k PPO updates. At 10k updates, TRACED requires $35.64 \pm 0.53$ h, about 7% more than ACCEL's $33.26 \pm 1.17$ h, yet achieves a 22% higher mean solved rate (Section 4).

Table 8: **Wall-Clock Training Time on BipedalWalker.** Average wall-clock duration for each algorithm

| Domain | TRACED 10k | ACCEL 10k | ACCEL 20k | ADD 10k | ADD 20k | PLR$^{\perp}$ 10k | PLR$^{\perp}$ 20k | DR 10k | DR 20k |
|---|---|---|---|---|---|---|---|---|---|
| BipedalWalker | $35.64 \pm 0.53$ | $33.26 \pm 1.17$ | $70.22 \pm 5.9$ | $21.13 \pm 0.11$ | $41.58 \pm 0.24$ | $35.49 \pm 1.07$ | $71.01 \pm 2.17$ | $18.38 \pm 0.1$ | $37.03 \pm 0.3$ |

## G.2    SAMPLE COMPLEXITY: ENVIRONMENT STEPS

Table 9 shows total environment steps required to achieve a fixed number of PPO updates. TRACED matches ACCEL's sample complexity exactly, confirming that its superior generalization stems from improved curriculum design rather than additional data.

Table 9: **Environment Steps.** Total environment interactions (millions) for each method per given number of student PPO updates.

| Environment | PPO Updates | PLR$^{\perp}$ | ADD | ACCEL | TRACED |
|---|---|---|---|---|---|
| MiniGrid | 10k | 82M | 41M | 93M | 93M |
| MiniGrid | 20k | 165M | 82M | 185M | – |
| BipedalWalker | 10k | 165M | 80M | 174M | 174M |
| BipedalWalker | 20k | 329M | 160M | 347M | – |

# H  OVERALL WORKFLOW

Algorithm 1 summarizes our overall UED procedure under the TRACED framework. We followed ACCEL (Parker-Holder et al., 2022) in all respects except the procedure used to compute task priority. At each iteration, with probability $1 - p_{\text{replay}}$ the teacher enters the *exploration phase* by sampling a fresh level $\theta \sim \mathcal{G}$, executing the current policy $\pi_\phi$ to collect a trajectory, and computing its approximated regret

$$\widehat{\text{Regret}}(\tau) = \text{PVL}(\tau) + \alpha \text{ATPL}(\tau).$$

If the buffer is full, we evict the lowest-priority task before appending. Otherwise, with probability $p_{\text{replay}}$ the teacher enters the *replay+mutation phase*, drawing a batch of $B$ tasks from the buffer proportional to their TaskPriority scores, updating the policy on each, and recomputing regret. We then select the $N_{\text{mutate}}$ tasks with the smallest regret, mutate each via the editor, evaluate the new variant, and replace its parent in the buffer. Finally, for every modified task we recompute TaskDifficulty, CoLearnability, and TaskPriority (Eq. 7) before proceeding to the next timestep.

---

**Algorithm 1** UED Workflow with Transition-Prediction Loss and Co-Learnability

---

1: **Given:** policy $\pi_\phi$, level generator $\mathcal{G}$, buffer capacity $K$, replay probability $p_{\text{replay}}$, batch size $B$, mutation count $N_{\text{mutate}}$, scaling factors $\alpha, \beta$
2: **Initialize** Task Buffer $\Lambda \leftarrow \emptyset$, timestep $t \leftarrow 0$
3: **while** policy has not converged **do**
4:     Sample decision $d_t \sim \text{Bernoulli}(p_{\text{replay}})$
5:     **if** $d_t = 0$ **then**                                          ▷ Exploration phase
6:         Sample new task $\theta \sim \mathcal{G}$
7:         Collect trajectory $\tau$ by executing $\pi_\phi$ on $\theta$ (no gradient)
8:         Compute regret estimate

$$\widehat{\text{Regret}}(\tau) = \text{PVL}(\tau) + \alpha \text{ATPL}(\tau)$$

9:         Append $\theta$ to buffer $\Lambda$ (if $|\Lambda| > K$, remove lowest-priority)
10:     **else**                                                         ▷ Replay + Mutation phase
11:         Sample a batch $\{\theta_k\}_{k=1}^B$ from $\Lambda$ w.p. $\propto \frac{1}{\text{TaskPriority}(\cdot, t)}$
12:         **for** $k = 1 \ldots B$ **do**
13:             Collect trajectory $\tau_k$ by executing $\pi_\phi$ on $\theta_k$
14:             Update policy $\phi$ using rewards from $\tau_k$
15:             Compute $\widehat{\text{Regret}}(\tau_k)$
16:         **end for**
17:         Select the $N_{\text{mutate}}$ tasks $\{\theta_{k'}\}$ with smallest $\widehat{\text{Regret}}(\tau_{k'})$
18:         **for** each selected $\theta_{k'}$ **do**
19:             $\tilde{\theta} \leftarrow \text{editor}(\theta_{k'})$
20:             Collect trajectory $\tilde{\tau}$ on $\tilde{\theta}$ (no gradient)
21:             Compute $\widehat{\text{Regret}}(\tilde{\tau})$
22:             Replace $\theta_{k'}$ in $\Lambda$ with $\tilde{\theta}$
23:         **end for**
24:     **end if**
25:     **Buffer update:**
26:     **for** each task $i \in \Lambda$ updated at $t$ **do**
27:         Recompute TaskDifficulty$(i, t)$
28:         Recompute CoLearnability$(i, t - 1)$
29:         Update TaskPriority$(i, t)$ via Eq. 7
30:     **end for**
31:     $t \leftarrow t + 1$
32: **end while**

---

# I IMPLEMENTATION DETAILS

## I.1 MINIGRID ENVIRONMENT

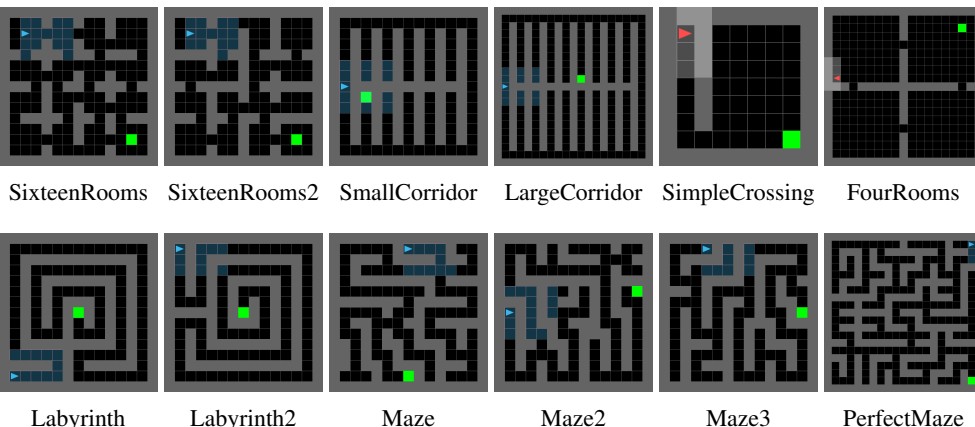

| SixteenRooms | SixteenRooms2 | SmallCorridor | LargeCorridor | SimpleCrossing | FourRooms |

| Labyrinth | Labyrinth2 | Maze | Maze2 | Maze3 | PerfectMaze |

Figure 19: **MiniGrid zero-shot environments used for evaluation.** In SmallCorridor and LargeCorridor, the goal can appear in any of the corridor paths. SimpleCrossing and FourRooms are adapted from MiniGrid (Chevalier-Boisvert et al., 2023), and PerfectMaze from **REPAIRED** (Jiang et al., 2021a).

The partially observable navigation environment is designed as a $15 \times 15$ grid maze, based on MiniGrid (Chevalier-Boisvert et al., 2023). Each tile in the maze can be either an empty tile, a wall, a goal, or the agent itself. The empty tile is a navigable space through which the agent can move, whereas the wall is an impassable obstacle that blocks the agent's movement. At the beginning of each episode, the initial position of the agent, the goal location, and the wall layout are randomly initialized. Up to 60 walls can be placed throughout the maze, increasing navigational complexity. At each timestep, the agent receives a $5 \times 5$ local observation, along with its current direction. The action space is defined as a discrete set of seven possible actions, although only three, turn left, turn right, and move forward, are relevant to maze navigation. These actions are selected without masking the unused action outputs. A reward is provided when the agent successfully reaches the goal, calculated as $1 - T/T_{\max}$, where $T$ is the number of steps taken and $T_{\max} = 250$ denotes the maximum allowed steps per episode. If the agent fails to reach the goal within the time limit, it does not receive a reward.

To evaluate performance in partially observable navigation, we include additional results on the MiniGrid environments, covering a suite of challenging zero-shot tasks from prior UED work (Dennis et al., 2020; Jiang et al., 2021a). Figure 19 displays all evaluation tasks.

## I.2 BIPEDALWALKER ENVIRONMENT

The environment is based on the BipedalWalkerHardcore environment of the OpenAI gym (Brockman et al., 2016). At each timestep, the agent receives a 24-dimensional observation, which includes information such as the hull angle and angular velocity, joint angles and speeds for the hips and knees, ground contact indicators for each foot, the robot's horizontal and vertical velocities, and lidar-based distance measurements to the ground ahead. The action space is represented by a 4-dimensional vector that controls the torques applied to the robot's two hip joints and two knee joints. Rewards are structured to encourage efficient and stable locomotion across the terrain. The agent receives positive rewards for processing while maintaining a straight hull posture, with small penalties applied for energy expenditure to discourage unnecessary motor use. The maximum achievable score is 300. If the agent falls or moves backward, the episode ends immediately, and a penalty of -100 is applied. The maximum episode length is 2000 timesteps. To introduce additional difficulty and test the robustness of the agent's locomotion policy, the environment includes a variety of challenging terrain features. These include **stairs**, which consist of sequences of elevated steps that the agent must ascend or descend; **pit gaps**, which are horizontal gaps over which the agent must

jump or step over; **stumps**, which are vertical obstacles with varying heights and widths that obstruct movement; and **surface roughness**, which introduces uneven ground textures that can disrupt balance and foot placement.

We evaluate agents in the BipedalWalker-v3 environment, which features relatively smooth terrain, as well as in the more demanding BipedalWalkerHardcore-v3, which introduces a variety of challenging obstacles. The BipedalWalker-v3 focuses on basic locomotion and balance without significant terrain disturbances. Terrain in this environment does not have obstacles such as stairs, pit gaps, and stumps, and only gentle slopes are generated by adding noise in the range of approximately -1 to 1. In contrast, BipedalWalkerHardcore-v3 includes complex terrain elements such as stairs, pit gaps, stumps, and roughness, which require precise control and adaptive strategies. These obstacles are procedurally generated with randomized parameters. The stair height is randomly set to either $+1$ (ascending) or $-1$ (descending), with the number of steps and step width randomly sampled between 3-5 steps and 4-5 units, respectively. The pit width is determined by sampling a single integer between 3-5 units. Both the height and width of each stump are sampled as a single random integer between 1-3 units, ensuring the stumps are always square-shaped. Terrain roughness noise range is the same as BipedalWalker-v3.

To further probe the agent's generalization capabilities, we include four targeted evaluation environments that isolate specific terrain challenges: Stairs, PitGap, Stump, and Roughness. Stairs specifies a fixed stair height of 2 units, with 5 steps and a step width between 4 and 5. PitGap sets the pit width to exactly 5 units. Stump defines the stump height as 2 units while allowing the stump width to vary between 1 and 2 units. Finally, Roughness generates terrain by adding noise 5. Figure 20 displays all evaluation tasks.

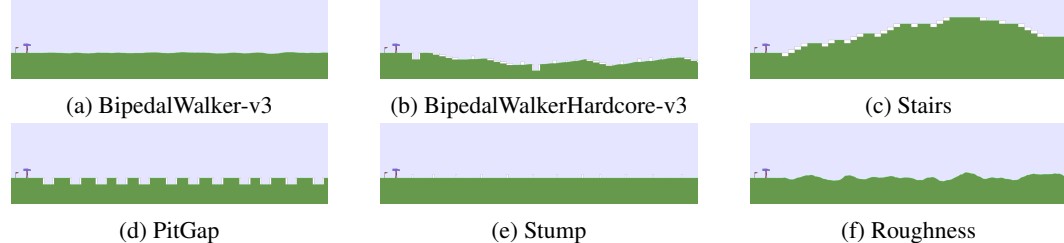

| (a) BipedalWalker-v3 | (b) BipedalWalkerHardcore-v3 | (c) Stairs |
| (d) PitGap | (e) Stump | (f) Roughness |

Figure 20: **Evaluation Terrains in BipedalWalker.** 6 held-out environments used to assess zero-shot generalization: (a) BipedalWalker-v3 with gentle noise-induced slopes; (b) BipedalWalkerHardcore-v3 combining multiple obstacle types; (c) Stairs, a fixed 5-step sequence requiring precise stepping; (d) PitGap, a 5-unit horizontal gap; (e) Stump, 2-unit tall obstacles with variable width; and (f) Roughness, continuous uneven ground generated by noise.

### I.3 NETWORK STRUCTURES

For partially observable navigation, the LSTM-based transition prediction model $f_\phi$ consists of three components: 1. an image encoder that compresses the input observation $s_t$ into a 128-dimensional latent embedding, 2. an LSTM module (hidden size 128, 1 layer) that processes the concatenation of this embedding and the action $a_t$, 3. and an image decoder that reconstructs the predicted next observation $\hat{s}_{t+1}$ from the LSTM output. We adopt an LSTM here because recurrent architectures have proven effective at capturing temporal dependencies in transition and video-prediction tasks under partial observability (Ha & Schmidhuber, 2018; Oh et al., 2015).

For walking in challenging terrain (BipedalWalker), the model instead concatenates the raw state vector $s_t$ with $a_t$ and feeds this into an LSTM (hidden size 128, 1 layer); the recurrent output is then linearly mapped to the predicted next state $\hat{s}_{t+1}$.

Its instantaneous error is defined by

$$\ell_t = \begin{cases} \|\hat{s}_{t+1} - s_{t+1}\|_1, & \text{in MiniGrid environments,} \\ \|\hat{s}_{t+1} - s_{t+1}\|_2^2, & \text{in BipedalWalker.} \end{cases}$$

We use L1 loss for MiniGrid because it preserves sharp edges and reduces blurring in image reconstruction (Zhao et al., 2016), while mean squared error is a standard, reliable choice for continuous state regression in locomotion domains (Mnih et al., 2015).

We adopt ACCEL's network architecture (Parker-Holder et al., 2022), differing only in the LSTM design choice. The PPO network serves as the student agent in our system, directly learning to optimize both the policy and value functions through interaction with the environment. The architecture follows the standard actor-critic structure where the actor head outputs an action distribution, and the critic head predicts the scalar value $V(s)$.

For partially observable navigation, the input consists of a 3-channel $5 \times 5$ local image observation and a directional scalar with dimension 4 indicating the agent's current orientation. The image is processed by a convolutional encoder with a filter size of 16, a kernel size of 3, and Rectified Linear Unit (ReLU) activations, followed by flattening. The directional scalar is passed through a fully connected layer of size 5, and the resulting embedding is concatenated with the flattened image features. The combined feature vector is fed separately to the actor and critic heads, each consisting of fully connected layers of size [32, 32]. The actor outputs a categorical distribution over discrete actions, while the critic produces a scalar value estimate $V(s)$.

For walking in a challenging terrain (BipedalWalker), the input is a flat state vector of dimension 24. This input is processed through a Multilayer Perceptron (MLP) with two hidden layers of 64 dimensions, followed by actor and critic heads. The actor head outputs the mean and standard deviation parameters of a diagonal Gaussian distribution over a continuous 4-dimensional action space, and the critic head outputs a scalar state value $V(s)$. The PPO learning process optimizes a clipped surrogate policy loss with clip parameter $\epsilon = 0.2$ combined with a value function loss.

## J  DIFFERENCE WITH FORMER STUDIES

### J.1  DIFFERENCE WITH ACCEL

Our approach builds on ACCEL's evolutionary, regret-based curriculum design, refining both its regret estimator and its sampling criterion. Like ACCEL, we maintain a level buffer and record each task's difficulty. However, instead of relying solely on Positive Value Loss (PVL) to approximate regret, we decompose regret into value error and transition-prediction error and incorporate both components into our difficulty measure.

Furthermore, prior UED methods sample levels independently based solely on their individual difficulty, ignoring cross-task transfer effects. We address this with our Co-Learnability metric, which quantifies how training on one task influences regret on others by measuring the average change in their difficulty after replay. By integrating Co-Learnability into our priority score, we can prioritize tasks that not only challenge the agent but also deliver broad transfer benefits across the entire task space.

### J.2  DIFFERENCE WITH CENIE

CENIE introduces a complementary curriculum strategy by explicitly measuring environmental novelty, rather than relying solely on regret signals as in $\mathrm{PLR}^{\perp}$ or ACCEL. It fits a Gaussian mixture model (GMM) to the agent's past trajectories and scores new tasks by how poorly the GMM explains them, thereby steering the agent toward unfamiliar regions of the state-action space. This model-agnostic approach can be applied on top of any UED framework, including PLR and ACCEL.

However, CENIE's diversity-driven objective differs from ours. TRACED's primary goal is to approximate task difficulty more faithfully, by decomposing regret into value and transition-prediction errors, and to capture how training on one task transfers to others via our Co-Learnability metric. While CENIE accounts for per-task novelty and regret, it ignores cross-task transfer effects. Moreover, TRACED is agnostic to the choice of level generator or editor, meaning our method could be combined with CENIE's novelty scoring to yield a curriculum that balances both transfer-aware difficulty and environmental diversity.

## K    RELATED WORKS

Unsupervised Environment Design (UED) defines a co-evolutionary framework in which a teacher agent generates task instances for a student policy, selecting those tasks with high learning potential to promote robust generalization to unseen environments (Wang et al., 2019; 2020; Dennis et al., 2020; Chung et al., 2024). PAIRED (Dennis et al., 2020) trains the teacher to maximize the regret between a protagonist and antagonist policy, causing the teacher to create increasingly challenging yet solvable environments. However, non-stationary task distributions and the high dimensionality of the task space impede convergence to optimal curricula. To mitigate these limitations, Mediratta et al. (2023) introduce entropy bonuses, alternative optimization algorithms, and online behavioral cloning, and Clutr (Azad et al., 2023) employs variational autoencoder-based unsupervised task representation learning to compress the task manifold and yield more stable curricula.

Another line of work is based on Prioritized Level Replay (PLR) (Jiang et al., 2021b), which avoids training a teacher agent and instead randomly generates tasks while introducing a replay buffer that stores previously generated tasks. This approach prioritizes tasks with high learning potential from the buffer to more effectively train the student agent. $\text{PLR}^{\perp}$ (Jiang et al., 2021a) make improvement of PLR, which used stop-gradient on trajectories from newly generated tasks but updated only with replayed tasks from buffer. This leads to better performance on unseen environments counterintuitively. ACCEL (Parker-Holder et al., 2022) extends the $\text{PLR}^{\perp}$ by implementing an evolutionary approach that makes small mutation to high-regret levels from the buffer. This evolutionary mechanism enables the development of progressively more complex challenges starting with simpler tasks. However, these methods (Jiang et al., 2021b;a; Parker-Holder et al., 2022) approximate regret solely via value loss, thereby implicitly learn true environment dynamics that could improve regret estimation.

Moving beyond regret-based approaches, CENIE (Teoh et al., 2024) integrates novelty with regret to expose agents to more diverse learning situations. Using Gaussian Mixture Models, CENIE quantifies how much a new task's trajectory differs from previous experiences stored in the buffer. This curriculum-awareness leads to better exploration and generalization. DIPLR (Li et al., 2023) introduces a diversity that quantifies the similarity between different levels by computing the Wasserstein distance between their occupancy distributions over state-action trajectories. MBeDED (Li et al., 2024) introduces the marginal benefit metric to quantify the performance gain of a student policy from training on a generated task. It compares a base policy (before training) and the updated student policy by measuring the difference in their expected returns on that task. SFL (Rutherford et al., 2024) assigns a learnability score to each level, computed from the empirical success rate achieved by evaluating the agent on sampled levels. Levels whose success rates fall in the intermediate range, neither too easy nor too difficult, are prioritized, guiding the agent toward tasks that are most conducive to further learning. A related idea appears in GoalGAN (Florensa et al., 2018), which is not a UED method but similarly adapts task difficulty by training a generative model to propose goals of suitable challenge. So far, to the best of our knowledge, there are no prior works that explicitly consider Co-Learnability between tasks.

UED's benefit on robust generalization performance in unseen environments has led to its application in other domains. For Multi-Agent Reinforcement Learning, MAESTRO (Samvelyan et al., 2023) extends UED to competitive multi-agent settings by creating a regret-based curriculum over the joint environment and co-player space, leveraging self-play to generate opponents while training agents robust to environment and co-player variations. In contrast, RACCOON (Erlebach & Cook, 2024) applies UED to cooperative scenarios, developing a curriculum that prioritizes high-regret partners from pre-trained partners to enhance collaboration. For meta-RL, GROOVE (Jackson et al., 2023) suggested policy meta-optimization (PMO) introducing algorithmic regret (AR), which measures the performance gap between a meta-learned optimizer and RL algorithms like A2C, creating curricula that identify informative environments for meta-training. DIVA (Costales & Nikolaidis, 2024) is the first method that extends UED to semi-supervised environment design, employing quality-diversity search to maintain a population of diverse training tasks in open-ended simulators, maximizing behavioral coverage. In sim-to-real settings, GCL (Wang et al., 2024) aligns the simulated curriculum with actual deployment tasks by sampling from real-world task distributions and adapting subsequent task generation based on the robot's past performance.

Similar to UED works, Curriculum Reinforcement Learning (CRL) accelerates agent training by constructing a sequence of progressively challenging tasks rather than directly confronting the agent with a single, complex target task (Narvekar et al., 2020). Early approaches distinguished between fixed curricula in which task order is predefined and self-paced learning, in which task selection adapts dynamically based on the agent's learning progress (Klink et al., 2020). Modern CRL methods have advanced this paradigm by framing curriculum generation as an inference process (Eimer et al., 2021; Klink et al., 2022). WAKER (Rigter et al., 2024) leverages reward-free curricula to train robust world models, thereby underscoring the versatility of self-paced approaches in diverse learning scenarios. Akin to the concept underlying SFL (Rutherford et al., 2024), ProCuRL (Tzannetos et al., 2023) operationalizes the pedagogical notion of the Zone of Proximal Development by using probability of success score-based scoring combined with softmax sampling to prioritize tasks of intermediate difficulty. However, these approaches assume access to the target task distribution, which conflicts with UED's core assumption of an unknown distribution.

## L    MORE DETAILS ON BASELINE ALGORITHMS

In Unsupervised Environment Design (UED), Domain Randomization (DR) constructs the curriculum by uniformly sampling tasks from the environment parameter space $\Theta$. Formally, DR samples tasks according to $\theta \sim p(\Theta)$ where $p(\Theta)$ represents a uniform distribution over the task parameter space. The agent is trained on these randomly sampled tasks.

Robust PLR (PLR$^\perp$) (Jiang et al., 2021a) was introduced as an extension of the original PLR (Jiang et al., 2021b). While DR treats all tasks equally, PLR$^\perp$ prioritizes tasks with high learning potential, maintaining a buffer $\Lambda$ of previously encountered tasks for replay. When sampling new tasks, the agent's policy parameters are stop-gradiented, meaning the policy is not updated on trajectories collected from these newly sampled tasks. The collected trajectories are used to compute PVL scores to decide whether to add the task to the replay buffer. Policy updates are exclusively performed on tasks sampled from the replay buffer. By stopping the gradient on randomly sampled levels, PLR$^\perp$ ensures that the policy is only updated on tasks specifically selected to maximize regret, leading to more robust generalization compared to both DR and PLR unintuitively.

Adversarially Compounding Complexity by Editing Levels (ACCEL) (Parker-Holder et al., 2022) actively evolves environments through an editing mechanism, allowing it to more efficiently explore the environment design space. This enables reusing the structure of sampled levels in the buffer for high-regret, rather than curating randomly sampled levels for high-regret. ACCEL maintains a buffer $\Lambda$ and employs a cycle of sampling, editing, and curation. The key insight of ACCEL is that regret serves as a domain-agnostic fitness function for evolution, enabling it to produce batches of levels at the frontier of agent capabilities. The editing mechanism involves making small mutations to previously high-regret levels, which can operate directly on environment elements such as blocks in MiniGrid. This evolutionary process creates an expanding frontier that matches the agent's capabilities, starting with simple levels and progressively increasing in complexity. Like PLR$^\perp$, ACCEL employs stop-gradient when evaluating new or edited levels to ensure theoretical guarantees.

Coverage-based Evaluation of Novelty In Environment (CENIE) (Teoh et al., 2024) introduces environment novelty as a complementary objective to regret in UED. CENIE quantifies environment novelty through state-action space coverage derived from the agent's accumulated experiences across previous environments in its curriculum. The framework operates on the intuition that a novel environment should induce unfamiliar experiences, pushing the student agent into unexplored regions of the state-action space. At the core of CENIE's implementation is the use of Gaussian Mixture Models (GMMs) to model the distribution of state-action pairs from the agent's past experiences. Given a state-action buffer $\Gamma$ containing pairs collected from previous environments, CENIE fits a GMM with parameters $\lambda_\Gamma$ to represent this distribution. The novelty of a candidate environment is then quantified by measuring the similarity between its newly observed state-action pairs and the learned distribution of past state-action experiences. CENIE integrates with existing UED algorithms by combining both novelty and regret into a unified priority score for environment selection.

Adversarial Environment Design via Regret-Guided Diffusion Models (ADD) (Chung et al., 2024) frames environment generation as a diffusion process steered by the agent's regret. Starting from uniformly sampled tasks, a pretrained score network $s_\phi$ is guided by the gradient of a differentiable regret estimate, computed via a learned return critic, during reverse diffusion.

In our empirical study, DR, PLR$^\perp$, and ACCEL were evaluated with the DCD codebase (Jiang et al., 2022), and assessed ADD with the ADD codebase (Chung et al., 2024). CENIE's performance data were incorporated from its original publication, as no open-source implementation was available.

## M   LIMITATIONS

As with any research, our approach presents several limitations that highlight opportunities for future investigation:

**Co-Learnability.** In this paper, we proposed Co-Learnability, simple yet effective method. Follow-up studies could address these gaps by developing multi-step Co-Learnability estimators, incorporating importance-weighting or counterfactual techniques to reduce sampling bias, and combining observational co-learn signals with auxiliary models (e.g., task-conditioned value predictors) for stronger causal inference.

**ATPL.** In environments with highly stochastic transitions, the ATPL signal (transition-prediction error) can become noisy, potentially reducing the fidelity of our regret approximation.

**Sampling levels.** We sampled a fixed number of the lowest-regret levels as easy tasks to mirror ACCEL's approach and enable fair comparisons with baseline methods. However, this simple strategy leaves room to explore more advanced task-selection mechanisms.

**Heuristic weighting.** We introduced weighted approach to regret approximation and task prioritization that relies on two key parameters $\alpha$, PVL and ATPL in the regret approximation, and $\beta$, which weights the relative importance of Co-Learnability against task difficulty. While these parameters were manually tuned in the current implementation and demonstrated effective results across environments, future work could explore more principled methods for automatically determining these weights.

**Extension to other RL algorithms.** Our experiments have been conducted with PPO-based student agents. While this allows for fair comparisons with previous work, there is a much broader range of algorithms that may interact differently with our regret-based curriculum. Off-policy methods such as SAC (Haarnoja et al., 2018) and TD3 (Fujimoto et al., 2018) feature distinct exploration strategies, which may yield different performance dynamics under our curriculum framework. Moreover, Model-based approaches like DreamerV3 (Hafner et al., 2025) build world models for planning, offering complementary insights into curriculum design. Evaluating these methods under our curriculum framework remains an important direction for future work.

**Extend to other UED methods.** We empirically demonstrated that TRACED yields more effective curricula, thereby enhancing agents' generalization capabilities at half the training cost. While our experiments focus on the ACCEL framework, future work will integrate our approach into alternative UED methods (PLR$^\perp$ (Jiang et al., 2021a) and CENIE (Teoh et al., 2024)) and conduct comprehensive experiments to assess its performance in these settings.

**Evaluation in other domains.**   Our experimental validation has been conducted in Mini-Grid (Chevalier-Boisvert et al., 2023; Dennis et al., 2020) and BipedalWalker (Brockman et al., 2016; Parker-Holder et al., 2022).   Evaluating TRACED in other domains such as Car Racing (Brockman et al., 2016; Jiang et al., 2021a), physical robotics applications (Mahmood et al., 2018), and reasoning task ARC (Chollet, 2019) would provide a more comprehensive assessment of the approach's generality and scalability across diverse settings.

## N   EXPERIMENTAL SETUP AND REPRODUCIBILITY

All experiments were conducted using Python 3.8.20 on Ubuntu 22.04.4 LTS. The hardware setup included an AMD EPYC 7543 32-core processor, 80GB of RAM, and an NVIDIA A100 GPU. Experiments were run on a shared server with 8 GPUs, with each experiment using a single GPU. We implemented all experiments in Python 3.8.20, using PyTorch(v1.9.0+cu111). The environment stack was built using MiniGrid (v1.0.1), Gym (v0.15.7).

## O  LLM USAGE

We used a large language model (LLM) solely for language editing. Concretely, the LLM assisted with grammar and style polishing, LaTeX phrasing (e.g., equation and caption wording), and improving clarity and concision of author-written text. The LLM was not used to generate ideas, design algorithms, select hyperparameters, run experiments, analyze data, create figures/tables, write code, or produce mathematical results.

## P  HYPERPARAMETERS

Table 10 summarizes the hyperparameters used. Unless otherwise noted, values are adopted directly from the DCD repository (Jiang et al., 2022). The only deviation from the DCD defaults is the number of PPO workers (see Appendix B.4). We tune two key weights, $\alpha$ and $\beta$, on MiniGrid and BipedalWalker, exploring $\alpha \in \{0.5, 1.0, 1.5\}$ and $\beta \in \{0.6, 0.8, 1.0\}$.

Table 10: **Hyperparameters used for training each method in each environment.**

| HyperParameter | MiniGrid | BipedalWalker |
|---|---|---|
| **PPO** | | |
| $\gamma$ | 0.995 | 0.99 |
| $\lambda_{\text{GAE}}$ | 0.95 | 0.9 |
| PPO rollout length | 256 | 2048 |
| PPO epochs | 5 | 5 |
| PPO minibatches per epoch | 1 | 32 |
| PPO clip range | 0.2 | 0.2 |
| PPO number of workers | 16 | 4 |
| Adam learning rate | 1e-4 | 3e-4 |
| Adam $\epsilon$ | 1e-5 | 1e-5 |
| PPO max gradient norm | 0.5 | 0.5 |
| PPO value clipping | True | False |
| Return normalization | False | True |
| Value loss coefficient | 0.5 | 0.5 |
| Student entropy coefficient | 0.0 | 1e-3 |
| Generator entropy coefficient | 0.0 | 1e-2 |
| **TRACED** | | |
| ATPL weight, $\alpha$ | 1.0 | 1.5 |
| Co-Learnability weight, $\beta$ | 1.0 | 0.6 |
| **ACCEL** | | |
| Replay rate, $p_{\text{replay}}$ | 0.8 | 0.9 |
| Buffer size, $K$ | 4000 | 1000 |
| Number of edits | 5 | 3 |
| Batch Size, $B$ | 4 | 4 |
| Number of mutated tasks, $N_{\text{mutate}}$ | 4 | 4 |
| Temperature | 0.3 | 0.1 |
| Staleness coefficient, $\rho$ | 0.3 | 0.5 |

# Q NUMERICAL RESULTS

## Q.1 MINIGRID ENVIRONMENT

Table 11: **MiniGrid Zero-Shot Solved Rates.** **Bold** indicates the best result per task; underline indicates the second-best.

| Environment | Update | DR | PLR$^\perp$ | ADD | ACCEL | TRACED |
|---|---|---|---|---|---|---|
| 16Rooms | 10k | $0.4 \pm 0.12$ | $0.81 \pm 0.06$ | $0.72 \pm 0.1$ | $0.76 \pm 0.12$ | $0.53 \pm 0.14$ |
| | 20k | $0.51 \pm 0.13$ | **$0.86 \pm 0.06$** | $0.64 \pm 0.12$ | $0.65 \pm 0.1$ | – |
| 16Rooms2 | 10k | $0.1 \pm 0.09$ | $0.42 \pm 0.11$ | $0.14 \pm 0.05$ | $0.48 \pm 0.13$ | **$0.58 \pm 0.13$** |
| | 20k | $0.27 \pm 0.11$ | $0.55 \pm 0.15$ | $0.18 \pm 0.1$ | **$0.58 \pm 0.14$** | – |
| SimpleCrossing | 10k | $0.58 \pm 0.05$ | $0.72 \pm 0.03$ | $0.63 \pm 0.03$ | $0.81 \pm 0.03$ | **$0.85 \pm 0.03$** |
| | 20k | $0.73 \pm 0.02$ | $0.8 \pm 0.03$ | $0.74 \pm 0.04$ | **$0.85 \pm 0.02$** | – |
| FourRooms | 10k | $0.32 \pm 0.04$ | $0.47 \pm 0.01$ | $0.39 \pm 0.03$ | $0.46 \pm 0.02$ | $0.46 \pm 0.02$ |
| | 20k | $0.5 \pm 0.02$ | $0.53 \pm 0.03$ | **$0.61 \pm 0.03$** | $0.49 \pm 0.03$ | – |
| SmallCorridor | 10k | $0.3 \pm 0.07$ | $0.52 \pm 0.13$ | $0.47 \pm 0.1$ | $0.37 \pm 0.14$ | $0.49 \pm 0.13$ |
| | 20k | **$0.74 \pm 0.1$** | $0.61 \pm 0.11$ | **$0.74 \pm 0.04$** | $0.22 \pm 0.13$ | – |
| LargeCorridor | 10k | $0.36 \pm 0.13$ | $0.33 \pm 0.13$ | $0.35 \pm 0.09$ | $0.44 \pm 0.15$ | $0.43 \pm 0.11$ |
| | 20k | $0.54 \pm 0.13$ | $0.32 \pm 0.11$ | **$0.72 \pm 0.06$** | $0.25 \pm 0.12$ | – |
| Labyrinth | 10k | $0.1 \pm 0.1$ | $0.5 \pm 0.15$ | $0.22 \pm 0.13$ | $0.6 \pm 0.15$ | **$0.93 \pm 0.03$** |
| | 20k | $0.58 \pm 0.16$ | $0.92 \pm 0.06$ | $0.9 \pm 0.1$ | $0.8 \pm 0.13$ | – |
| Labyrinth2 | 10k | $0.1 \pm 0.1$ | $0.28 \pm 0.13$ | $0.0 \pm 0.0$ | $0.52 \pm 0.14$ | **$0.96 \pm 0.03$** |
| | 20k | $0.18 \pm 0.12$ | $0.64 \pm 0.15$ | $0.63 \pm 0.14$ | $0.78 \pm 0.13$ | – |
| Maze | 10k | $0.02 \pm 0.01$ | $0.13 \pm 0.1$ | $0.04 \pm 0.02$ | $0.41 \pm 0.13$ | **$0.7 \pm 0.1$** |
| | 20k | $0.1 \pm 0.1$ | $0.52 \pm 0.16$ | $0.2 \pm 0.13$ | $0.59 \pm 0.14$ | – |
| Maze2 | 10k | $0.19 \pm 0.12$ | $0.43 \pm 0.14$ | $0.09 \pm 0.08$ | $0.44 \pm 0.15$ | $0.6 \pm 0.1$ |
| | 20k | $0.16 \pm 0.11$ | $0.56 \pm 0.14$ | $0.28 \pm 0.1$ | **$0.65 \pm 0.13$** | – |
| Maze3 | 10k | $0.27 \pm 0.13$ | $0.57 \pm 0.13$ | $0.2 \pm 0.1$ | $0.78 \pm 0.13$ | $0.81 \pm 0.06$ |
| | 20k | $0.46 \pm 0.14$ | **$0.95 \pm 0.04$** | $0.54 \pm 0.16$ | $0.86 \pm 0.09$ | – |
| PerfectMaze(M) | 10k | $0.06 \pm 0.02$ | $0.24 \pm 0.05$ | $0.17 \pm 0.08$ | $0.41 \pm 0.05$ | **$0.64 \pm 0.07$** |
| | 20k | $0.2 \pm 0.05$ | $0.55 \pm 0.08$ | $0.36 \pm 0.08$ | $0.53 \pm 0.08$ | – |
| Mean | 10k | $0.23 \pm 0.05$ | $0.45 \pm 0.04$ | $0.28 \pm 0.04$ | $0.54 \pm 0.03$ | **$0.66 \pm 0.03$** |
| | 20k | $0.41 \pm 0.03$ | $0.65 \pm 0.04$ | $0.55 \pm 0.04$ | $0.6 \pm 0.05$ | – |

Table 11 reports per-task zero-shot solved rates on twelve held-out MiniGrid mazes, averaged over 10 seeds (mean ± s.e.), following Agarwal et al. (2021). We compare TRACED at 10k updates against DR, PLR$^\perp$, ACCEL, and ADD (each at 10k and 20k). At 10k, TRACED achieves the highest solved rate in six mazes (16Rooms2, SimpleCrossing, Labyrinth, Labyrinth2, Maze, PerfectMaze(M)) and the second-best in one maze (Maze2). Averaged across all twelve mazes, TRACED attains a mean solved rate of 0.66, outperforming all baseline configurations.

Table 12: **MiniGrid Ablation Results.** **Bold** indicates the best result per task; underline indicates the second-best.

| Environment | ACCEL 10k | TRACED - ATPL 10k | TRACED - CL 10k | TRACED 10k |
|---|---|---|---|---|
| 16Rooms | $0.7 \pm 0.19$ | $0.72 \pm 0.14$ | **$0.91 \pm 0.06$** | $0.79 \pm 0.19$ |
| 16Rooms2 | $0.44 \pm 0.17$ | $0.01 \pm 0.01$ | **$0.79 \pm 0.17$** | $0.72 \pm 0.17$ |
| SimpleCrossing | $0.77 \pm 0.02$ | $0.68 \pm 0.04$ | $0.86 \pm 0.04$ | **$0.89 \pm 0.01$** |
| FourRooms | $0.46 \pm 0.04$ | $0.36 \pm 0.02$ | **$0.48 \pm 0.04$** | $0.47 \pm 0.02$ |
| SmallCorridor | $0.03 \pm 0.03$ | $0.44 \pm 0.21$ | **$0.57 \pm 0.2$** | $0.49 \pm 0.17$ |
| LargeCorridor | $0.11 \pm 0.1$ | $0.42 \pm 0.22$ | **$0.65 \pm 0.21$** | $0.5 \pm 0.14$ |
| Labyrinth | $0.55 \pm 0.23$ | $0.4 \pm 0.24$ | $0.49 \pm 0.2$ | **$1.0 \pm 0.0$** |
| Labyrinth2 | $0.88 \pm 0.1$ | $0.16 \pm 0.11$ | $0.62 \pm 0.18$ | **$0.98 \pm 0.01$** |
| Maze | $0.42 \pm 0.24$ | $0.29 \pm 0.19$ | **$0.63 \pm 0.2$** | $0.59 \pm 0.17$ |
| Maze2 | **$0.42 \pm 0.24$** | $0.3 \pm 0.19$ | $0.41 \pm 0.15$ | **$0.42 \pm 0.19$** |
| Maze3 | $0.8 \pm 0.2$ | $0.68 \pm 0.2$ | **$0.86 \pm 0.12$** | **$0.86 \pm 0.07$** |
| PerfectMaze(M) | $0.33 \pm 0.04$ | $0.34 \pm 0.02$ | $0.53 \pm 0.07$ | **$0.66 \pm 0.11$** |
| **Mean** | $0.49 \pm 0.04$ | $0.4 \pm 0.04$ | $0.65 \pm 0.03$ | **$0.7 \pm 0.04$** |

Table 12 presents zero-shot solved rates on twelve held-out MiniGrid mazes for TRACED and its ablations at 10k updates, averaged over 10 seeds (mean ± s.e.). Full TRACED achieves the highest score in six mazes (SimpleCrossing, Labyrinth, Labyrinth2, Maze3, PerfectMaze(M), and Four-Rooms) and ranks second in the remaining six, yielding a mean solved rate of 0.70 and outperforming every variant. These results indicate that both the transition-prediction error and Co-Learnability are critical to TRACED's strong, consistent performance across diverse tasks.

Table 13: **PerfectMaze Zero-Shot Solved Rates.** **Bold** indicates the best result per environment; underline indicates the second-best.

| Environment | Update | DR | PLR$^\perp$ | ADD | ACCEL | TRACED |
|---|---|---|---|---|---|---|
| PerfectMaze(Large) | 10k | $0.0 \pm 0.0$ | $0.02 \pm 0.01$ | $0.04 \pm 0.04$ | $0.09 \pm 0.03$ | $\mathbf{0.27 \pm 0.07}$ |
| | 20k | $0.02 \pm 0.01$ | $0.19 \pm 0.09$ | $0.18 \pm 0.1$ | $\underline{0.20} \pm 0.08$ | – |
| PerfectMaze(XL) | 10k | $0.0 \pm 0.0$ | $0.01 \pm 0.0$ | $0.02 \pm 0.02$ | $0.01 \pm 0.01$ | $0.1 \pm 0.04$ |
| | 20k | $0.0 \pm 0.0$ | $\underline{0.12} \pm 0.09$ | $\mathbf{0.14} \pm 0.09$ | $0.09 \pm 0.06$ | – |

Table 13 compares zero-shot solved rates of TRACED against DR, PLR$^\perp$, ADD, and ACCEL on two large-scale PerfectMaze environments: PerfectMazeLarge and PerfectMazeXL, averaged over 10 seeds (mean ± s.e.). After 10k updates, TRACED achieves the highest success rate on Perfect-MazeLarge (27% ± 7%) and the third-highest on PerfectMazeXL (10% ± 4%), matching ADD at 20k. These results underscore TRACED's ability to scale to extreme maze sizes.

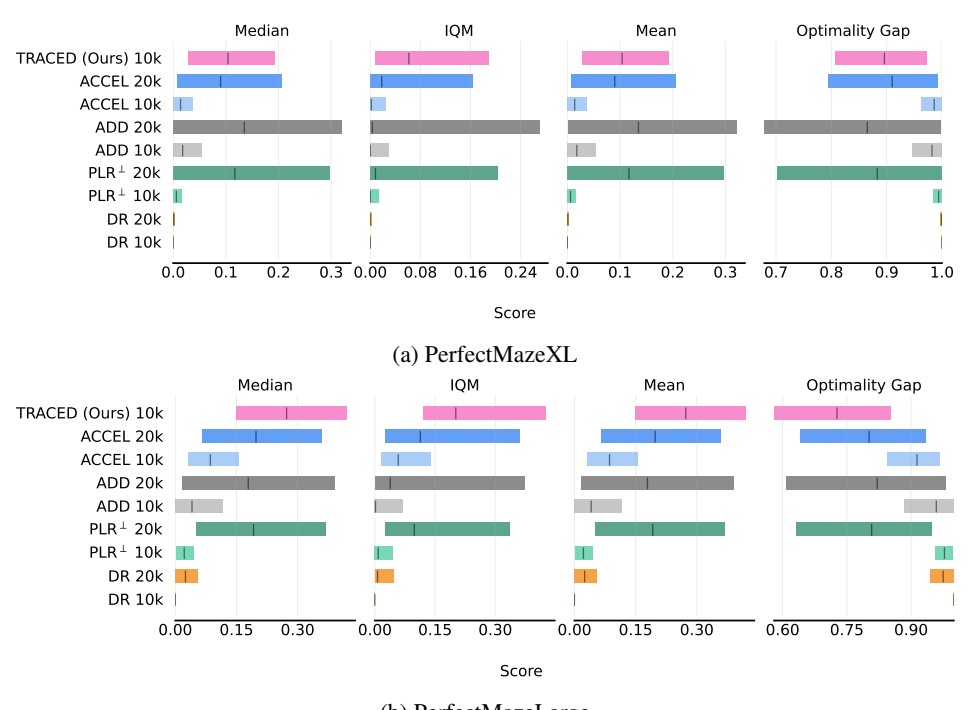

(a) PerfectMazeXL

(b) PerfectMazeLarge

Figure 21: **Aggregate zero-shot performance on PerfectMaze.** Full metric curves (Median, IQM, Mean, Optimality Gap) for PerfectMazeXL (top) and PerfectMazeLarge (bottom).

Figure 21 presents the full aggregate metrics for both PerfectMaze benchmarks. Despite the high variance inherent to these large-scale tasks, TRACED achieves the strongest 10k performance and closely matches the best 20k performance among all baselines.

## Q.2 BIPEDALWALKER ENVIRONMENT

Table 14 compares TRACED's zero-shot returns with DR, PLR$^\perp$, ADD, ACCEL, and ACCEL-CENIE on six BipedalWalker terrains, averaged over five seeds (mean $\pm$ s.e.). ACCEL-CENIE is estimated by applying the ACCEL-to-CENIE performance ratio reported in Teoh et al. (2024) to our ACCEL implementation. At 10k updates, TRACED achieves the highest mean return on three terrains (Basic, Hardcore, and Stump) and attains an overall mean of $89.95 \pm 31.72$, outperforming all baselines.

Table 14: **BipedalWalker Zero-Shot Test Returns.** **Bold** indicates the best result per terrain; underline indicates the second-best.

| Environment | Update | DR | PLR$^\perp$ | ADD | ACCEL | ACCEL-CENIE | TRACED |
|---|---|---|---|---|---|---|---|
| Basic | 10k | $112.56 \pm 62.99$ | $131.63 \pm 56.58$ | $119.43 \pm 63.72$ | $281.65 \pm 5.25$ | $273.56 \pm 1.5$ | **293.67** $\pm 3.56$ |
| | 20k | $68.17 \pm 44.71$ | $97.67 \pm 61.73$ | $63.64 \pm 62.71$ | $\underline{281.85} \pm 3.72$ | $275.04 \pm 3.19$ | – |
| Hardcore | 10k | $-19.67 \pm 7.35$ | $-17.36 \pm 5.99$ | $3.83 \pm 9.07$ | $37.59 \pm 15.0$ | $66.83 \pm 17.48$ | **86.83** $\pm 17.96$ |
| | 20k | $-16.98 \pm 5.8$ | $-23.7 \pm 2.05$ | $10.48 \pm 20.12$ | $59.23 \pm 25.5$ | $\underline{84.09} \pm 34.32$ | – |
| Stairs | 10k | $-17.6 \pm 7.21$ | $\underline{-7.23} \pm 7.16$ | **-5.87** $\pm 9.07$ | $-38.71 \pm 10.54$ | $-30.4 \pm 8.35$ | $-29.0 \pm 10.4$ |
| | 20k | $-9.11 \pm 7.67$ | $-10.2 \pm 4.5$ | $-0.66 \pm 9.2$ | $-46.34 \pm 5.78$ | $-36.39 \pm 11.48$ | – |
| PitGap | 10k | $-34.39 \pm 15.66$ | $-46.21 \pm 15.69$ | $-17.8 \pm 13.82$ | $-65.07 \pm 7.57$ | $-39.81 \pm 18.75$ | $-39.26 \pm 11.42$ |
| | 20k | $\underline{-26.03} \pm 10.98$ | $-48.72 \pm 12.23$ | **-7.65** $\pm 7.52$ | $-64.89 \pm 18.93$ | $-45.92 \pm 33.9$ | – |
| Stump | 10k | $-24.99 \pm 6.76$ | $-27.74 \pm 6.33$ | $-26.15 \pm 8.49$ | $-79.18 \pm 5.45$ | $-60.05 \pm 10.96$ | **34.16** $\pm 54.58$ |
| | 20k | $-25.52 \pm 12.39$ | $-23.61 \pm 2.86$ | $\underline{3.26} \pm 19.1$ | $-67.18 \pm 15.56$ | $-46.34 \pm 176.21$ | – |
| Roughness | 10k | $-0.66 \pm 12.01$ | $3.55 \pm 9.91$ | $20.61 \pm 13.78$ | $161.72 \pm 28.36$ | $174.92 \pm 2.25$ | $193.29 \pm 21.6$ |
| | 20k | $0.19 \pm 8.18$ | $-2.73 \pm 9.51$ | $18.71 \pm 21.59$ | $\underline{213.48} \pm 7.69$ | **224.4** $\pm 8.63$ | – |
| **Mean** | 10k | $2.54 \pm 15.44$ | $6.1 \pm 10.69$ | $14.4 \pm 14.05$ | $49.67 \pm 11.24$ | $64.18$ | **89.95** $\pm 12.95$ |
| | 20k | $-1.55 \pm 8.83$ | $-1.89 \pm 8.93$ | $14.63 \pm 21.99$ | $62.69 \pm 8.91$ | $\underline{75.81}$ | – |

Table 15 reports zero-shot solved rates across six BipedalWalker terrains for DR, PLR$^\perp$, ADD, ACCEL, and TRACED, averaged over five seeds. A trajectory is considered "solved" if its return exceeds 230. At 10k updates, TRACED achieves the highest solved rate on five of the six tasks (Basic, Hardcore, Stairs, PitGap, Stump) and ranks second on Roughness, yielding an overall mean of 0.36, substantially higher than ACCEL's 0.29 and the other baselines. No baseline matches TRACED's combined efficiency and reliability in zero-shot generalization.

Table 15: **BipedalWalker Zero-Shot Solved Rates.** **Bold** indicates the best per terrain; underline indicates the second-best.

| Environment | Update | DR | PLR$^\perp$ | ADD | ACCEL | TRACED |
|---|---|---|---|---|---|---|
| Basic | 10k | $0.35 \pm 0.2$ | $0.43 \pm 0.21$ | $0.34 \pm 0.21$ | $0.98 \pm 0.02$ | **1.00** $\pm 0.0$ |
| | 20k | $0.16 \pm 0.13$ | $0.35 \pm 0.22$ | $0.2 \pm 0.2$ | $\underline{0.99} \pm 0.0$ | – |
| Hardcore | 10k | $0.0 \pm 0.0$ | $0.0 \pm 0.0$ | $0.01 \pm 0.01$ | $0.15 \pm 0.03$ | **0.28** $\pm 0.05$ |
| | 20k | $0.0 \pm 0.0$ | $0.0 \pm 0.0$ | $0.04 \pm 0.04$ | $\underline{0.23} \pm 0.08$ | – |
| Stairs | 10k | $0.0 \pm 0.0$ | $0.0 \pm 0.0$ | $0.0 \pm 0.0$ | **0.03** $\pm 0.02$ | **0.03** $\pm 0.02$ |
| | 20k | $0.0 \pm 0.0$ | $0.0 \pm 0.0$ | $0.0 \pm 0.0$ | **0.03** $\pm 0.01$ | – |
| PitGap | 10k | $0.0 \pm 0.0$ | $0.0 \pm 0.0$ | $0.0 \pm 0.0$ | $0.0 \pm 0.0$ | **0.02** $\pm 0.01$ |
| | 20k | $0.0 \pm 0.0$ | $0.0 \pm 0.0$ | $0.0 \pm 0.0$ | **0.02** $\pm 0.02$ | – |
| Stump | 10k | $0.0 \pm 0.0$ | $0.0 \pm 0.0$ | $0.0 \pm 0.0$ | $0.0 \pm 0.0$ | **0.18** $\pm 0.12$ |
| | 20k | $0.0 \pm 0.0$ | $0.0 \pm 0.0$ | $\underline{0.02} \pm 0.02$ | $0.0 \pm 0.0$ | – |
| Roughness | 10k | $0.01 \pm 0.01$ | $0.0 \pm 0.01$ | $0.01 \pm 0.01$ | $0.56 \pm 0.1$ | $\underline{0.65} \pm 0.08$ |
| | 20k | $0.0 \pm 0.0$ | $0.0 \pm 0.0$ | $0.03 \pm 0.03$ | **0.76** $\pm 0.03$ | – |
| **Mean** | 10k | $0.06 \pm 0.04$ | $0.07 \pm 0.04$ | $0.06 \pm 0.04$ | $0.29 \pm 0.02$ | **0.36** $\pm 0.03$ |
| | 20k | $0.03 \pm 0.02$ | $0.06 \pm 0.04$ | $0.05 \pm 0.05$ | $\underline{0.34} \pm 0.03$ | – |

# R    COMPARISON WITH SFL

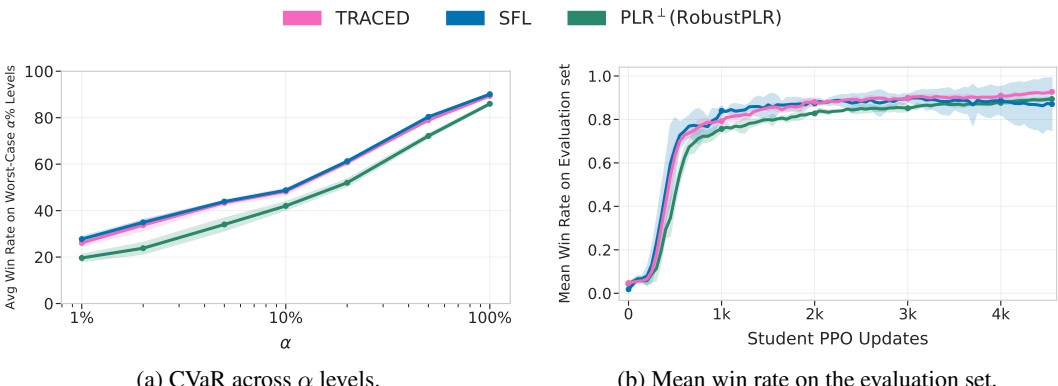

(a) CVaR across $\alpha$ levels.    (b) Mean win rate on the evaluation set.

Figure 22: **TRACED vs. SFL and Robust PLR on multi-agent JaxNav.** TRACED consistently outperforms Robust PLR and achieves performance comparable to SFL across all metrics.

Sampling for Learnability (SFL) (Rutherford et al., 2024) is a learnability-driven UED method that prioritizes levels with intermediate difficulty by selecting tasks with high learnability scores, computed as $p(1-p)$ where $p$ is the agent's empirical success rate on a given level obtained from rollouts.

We evaluate TRACED on the multi-agent JaxNav benchmark following the evaluation protocol introduced in Rutherford et al. (2024), including CVaR and mean win rate computed over sampled evaluation levels. While the original SFL experiments trained baselines for 22,850 PPO updates, we perform our comparison using 4,550 updates for computational efficiency. This training regime corresponds to a phase in which SFL is reported to exhibit strong performance relative to other UED baselines (Rutherford et al., 2024). Aside from the inclusion of TRACED, all training and evaluation details (e.g., level sampling, CVaR estimation, and construction of evaluation sets) match those in Rutherford et al. (2024). Results are averaged over three seeds.

Figure 22 shows CVaR across several risk levels and the mean win rate on the evaluation set. In addition to SFL, we include PLR$^\perp$ (Robust PLR), the strongest baseline used in Rutherford et al. (2024). TRACED consistently outperforms Robust PLR across all CVaR levels and achieves performance comparable to SFL in both CVaR and mean win rate. Notably, TRACED attains this performance without relying on any learnability-driven evaluation during training.

These results suggest that TRACED captures many of the benefits of learnability-based curricula while depending solely on a regret-based difficulty signal. Incorporating SFL-style evaluation tools, such as those used to quantify Co-Learnability, may further enhance TRACED's performance in complex multi-agent settings.

## S   TEMPORAL CONSISTENCY OF CO-LEARNABILITY

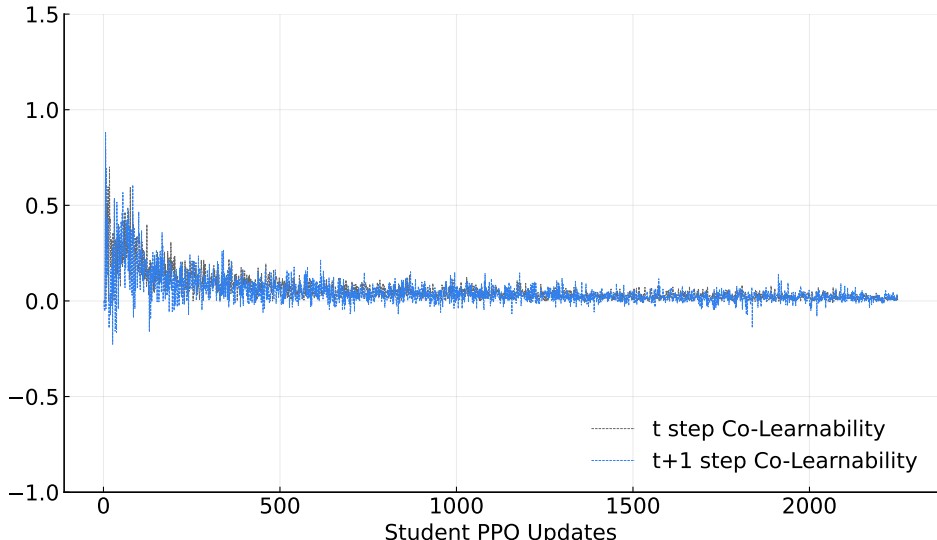

Figure 23: **Temporal consistency of Co-Learnability.**  Shadow testing produces aligned Co-Learnability sequences, enabling direct comparison of $\mathrm{CoLearnability}_i(t)$ and $\mathrm{CoLearnability}_i(t+1)$. The two signals exhibit positive correlation throughout training.

To examine whether training on tasks with high Co-Learnability at time $t$ continues to facilitate progress at time $t+1$, we quantify the temporal consistency of Co-Learnability values. A key challenge is that Co-Learnability is only computed for tasks selected by the curriculum at each step, resulting in irregularly sampled time series in which many tasks are not evaluated at both $t$ and $t+1$. This prevents direct computation of temporal correlation.

To obtain time-aligned Co-Learnability estimates, we apply a shadow-testing procedure. Let $L_t$ denote the indices of levels selected by the curriculum at timestep $t$. After the update on $L_t$, we clone the agent's parameters to construct a shadow agent. While the main learner proceeds normally (so that $L_{t+1}$ may differ from $L_t$), the shadow agent is forced to replay the same levels by setting $L_{t+1} = L_t$. At the next timestep, the shadow agent resumes standard TRACED updates without further intervention. This procedure yields aligned pairs of Co-Learnability values, $\mathrm{CoLearnability}_i(t)$ and $\mathrm{CoLearnability}_i(t+1)$, for the same set of levels across all timesteps, enabling direct temporal comparison.

Figure 23 shows the resulting time series of Co-Learnability at steps $t$ and $t+1$, averaged over three random seeds on multi-agent JaxNav. Although both signals exhibit substantial variance early in training, they converge toward small positive values as the policy improves and performance becomes more uniform across levels. The temporal correlation between the two estimates is summarized below:

- Pearson correlation: $0.3998$ ($p \approx 3.9 \times 10^{-87}$)
- Spearman correlation: $0.2013$ ($p \approx 5.4 \times 10^{-22}$)
- Time-averaged Co-Learnability at step $t$: $0.0547$
- Time-averaged Co-Learnability at step $t+1$: $0.0473$

The slightly lower mean at step $t+1$ is expected, as Co-Learnability values naturally decrease over training due to overall improvement across levels. The positive correlations indicate that tasks exhibiting higher Co-Learnability at time $t$ tend to retain elevated Co-Learnability at $t+1$, suggesting mild but non-negligible temporal persistency.

