# OpenReview forum: "TRACED: Transition-aware Regret Approximation with Co-learnability for Environment Design"
_ICLR.cc/2026/Conference — ICLR 2026 Poster_

### Official Review · Reviewer_mbPu · 2025-10-30

**Soundness:** 3
**Presentation:** 3
**Contribution:** 2
**Rating:** 4
**Confidence:** 4

**Summary:**

This paper focuses on the problem of Unsupervised Environment Design (UED)  by improving the environment sampling strategy. For instance, it enhances the regret estimation metric by incorporating the model's transition-prediction error , and proposes a 'Co-Learnability' metric. These contributions are designed to efficiently sample environment tasks with high learning potential , and the experiments demonstrate that this method effectively improves efficiency.

**Strengths:**

1. The paper's core insight is introduced very clearly, allowing the reader to easily follow the authors' thought process and understand the proposed method. The writing quality is high.

2. The concept of "Co-Learnability" is novel and interesting, and the paper provides a helpful visual analysis to support it . This idea appears to be a promising direction for multi-task generalization in reinforcement learning and deserves further investigation.

3. The experimental analysis is both thorough and solid, supported by sufficient ablation studies and insightful visualizations.

**Weaknesses:**

1. The novelty of the contributions appears limited. The use of a transition-prediction (reconstruction) loss is a well-established technique in RL, and the method itself is an incremental improvement upon the existing frameworks.

2. The estimation of 'Co-Learnability' is relatively coarse. As it is highly dependent on the sampling of the batch $\mathcal{T}$, its stability is a concern. The justification for using ATPL as a proxy for the 'Future value gap' (term iii in Eq. 2) lacks sufficient theoretical backing or direct experimental validation; it appears to be a heuristic choice rather than a formally derived approximation.

3. The improvement in performance appears relatively limited, even though the sample efficiency has increased. Moreover, the experiments are conducted in relatively simple envs (MiniGrid and Bipedal Walker) , which calls into question the practical to more complex tasks .

**Questions:**

1. Although the authors attempt to use Equation (2) to illustrate their insight for introducing ATPL, there is no theoretical proof of the relationship between ATPL and term (iii) . I believe it would be more convincing if the paper could show the progression curve of regret during training, both with and without the ATPL component.

2. I have concern about the reconstruction loss. The performance improvement from ATPL stems from introducing dynamics modeling into the loop. Is it possible that introducing the dynamics model directly on the student agent (e.g., as an auxiliary loss) would be more direct and potentially yield even better results?

---

> ### Author Response · Authors · 2025-11-24
>
> We would like to express our sincere gratitude for your detailed and thoughtful feedback. We respond to each point below.
>
> ### Questions
>
> > Q1. Although the authors attempt to use Equation (2) to illustrate their insight for introducing ATPL, there is no theoretical proof of the relationship between ATPL and term (iii) . I believe it would be more convincing if the paper could show the progression curve of regret during training, both with and without the ATPL component.
>
> Computing ground-truth regret requires access to the optimal value function, which is unavailable in practical environments. For this reason, plotting a direct regret–ATPL progression curve is not feasible. Instead, we provide a theoretically grounded analysis in Appendix R, where we show that incorporating ATPL into the regret approximation yields a strictly tighter upper bound on the regret mismatch, thereby producing a more faithful surrogate for true regret.
>
> > Q2. I have concern about the reconstruction loss. The performance improvement from ATPL stems from introducing dynamics modeling into the loop. Is it possible that introducing the dynamics model directly on the student agent (e.g., as an auxiliary loss) would be more direct and potentially yield even better results?
>
> The LSTM used to compute ATPL is implemented as a separate transition-prediction backbone, and does not share weights with the actor–critic network. Although parameter sharing is technically possible, it couples the agent’s policy-optimization objective with transition-prediction accuracy. This risks gradient interference and makes it difficult to isolate the effect of ATPL on curriculum quality. To allow a clean assessment of ATPL’s utility as a regret-based correction term, we intentionally use a distinct transition model.
>
> ### Additional Analyses
>
> We conducted new experiments on the JaxNav-MultiAgent benchmark used in SFL, comparing TRACED, SFL, and Robust PLR. Following the exact SFL evaluation protocol (CVaR and mean win rate), we compare TRACED, SFL, and Robust PLR. Results are included in Appendix Q. TRACED consistently outperforms Robust PLR across all CVaR levels and achieves performance comparable to SFL. The full evaluation code for JaxNav-MultiAgent has been added to the supplementary materials.
>
> We thank the reviewer once again for the thoughtful comments, which helped us strengthen our theoretical justification and expand our empirical evaluation. We hope our clarifications adequately address the concerns and would appreciate the reviewer’s consideration of our responses in their overall assessment.

---

> > ### Comment · Reviewer_mbPu · 2025-11-25
> >
> > Thank you for your response and clarifications. While I appreciate the explanation regarding the design choices, I remain unconvinced about the effectiveness and necessity of the proposed components due to the limited empirical evidence provided.
> >
> > As my main concerns regarding the method's soundness and contribution remain, I will keep my current rating.

---

> > > ### Author Response · Authors · 2025-11-27
> > >
> > > Thank you for the thoughtful follow-up and for taking the time to reassess our work. We appreciate your careful review and constructive feedback throughout the process.

---

### Official Review · Reviewer_y3M6 · 2025-11-01

**Soundness:** 3
**Presentation:** 3
**Contribution:** 3
**Rating:** 6
**Confidence:** 4

**Summary:**

The paper studies the unsupervised environment design (UED) problem, where a teacher adaptively generates tasks and a student learns a policy from them. The authors identify a limitation in existing UED methods: their learning potential (task difficulty) measures fail to capture the mismatch between learned and true dynamics when approximating regret. To address this, the authors refine regret estimation by augmenting the positive value loss with a transition-prediction error term. Based on the refined regret and a co-learnability metric (measuring how training on one task affects others), they propose a new curriculum strategy called TRACED. Empirical results on MiniGrid and BipedalWalker demonstrate the efficacy of TRACED, supported by thorough ablation studies and wall-clock time comparisons.

**Strengths:**

The paper is generally well written and provides a solid review of related work.

Proposes a novel and computationally efficient curriculum strategy for UED that achieves competitive performance without additional overhead.

Includes systematic empirical evaluation, with detailed ablation studies and reporting of wall-clock time.

**Weaknesses:**

Clarity of Section 3.2 could be improved:
- co-learnability term in Eq. (7) requires one-step lookahead in the curriculum
- task-difficulty $(i, t)$ defined as the regret approximation of the task $i$ when last sampled before time $t$
- co-learnability $(i, t)$ computed when task $i$ was last drawn before time $t-1$
- Please clarify why the rank transform prevents outliers from dominating the sampling distribution.

Missing discussion of ZPD-based curriculum strategies (e.g., Florensa et al., 2018; Tzannetos et al., 2023).

SFL was excluded as a baseline due to a lack of advantage in MiniGrid, but in other environments, SFL has been reported to outperform PLR. Are experiments on those domains prohibitively expensive (if so, this is understandable)?

Florensa et al., 2018: Automatic Goal Generation for Reinforcement Learning Agents.

Tzannetos et al., 2023: Proximal Curriculum for Reinforcement Learning Agents.

**Questions:**

clarification question: the learned policy $\pi_\phi$ is not conditioned task/environment parameter $\theta$, but rather adapts through trajectory conditioning (i.e., non-stationary behavior), right?

Lines 268-269 mention post-convergence oscillations in long-horizon MiniGrid runs for TRACED and ACCEL. How is early stopping handled in practice-by evaluating on a validation set and stopping when returns plateau?

---

> ### Author Response · Authors · 2025-11-24
>
> We sincerely appreciate the reviewer’s careful reading and insightful comments. We address each point in detail below.
>
> ### Weaknesses
>
> We have expanded Section 3.2 to address all clarity points raised by the reviewer. We have also added a discussion of ZPD-based curriculum strategies (Florensa et al., 2018; Tzannetos et al., 2023) to the Related Work section, highlighting conceptual connections.
>
> To directly address the reviewer’s concern, we conducted additional experiments on the JaxNav-MultiAgent benchmark used in SFL, comparing TRACED, SFL, and Robust PLR under the SFL evaluation protocol (CVaR and mean win rate). The full results are provided in Appendix Q. Across all CVaR levels, TRACED consistently outperforms Robust PLR and achieves performance comparable to SFL. The full JaxNav-MultiAgent evaluation code has been added to the supplementary materials.
>
> ### Questions
>
> > Q1. clarification question: the learned policy $\pi_\phi$ is not conditioned task/environment parameter $\theta$, but rather adapts through trajectory conditioning (i.e., non-stationary behavior), right?
>
> Yes, the policy is not conditioned on task/environment parameters. The agent instead adapts through trajectory conditioning.
>
> > Q2. Lines 268-269 mention post-convergence oscillations in long-horizon MiniGrid runs for TRACED and ACCEL. How is early stopping handled in practice-by evaluating on a validation set and stopping when returns plateau?
>
> UED aims to achieve strong performance across all tasks, making it inherently difficult to define a principled validation set because the underlying target distribution is unknown. In practice, one could construct a small, carefully selected held-out set that reasonably reflects the task distribution and use it for early stopping.
>
> Importantly, our choice of reporting results at 10k updates is an evaluation point determined before any oscillatory behavior emerges, as shown in the long-term analysis in Appendix P. Across seeds and methods, performance generally saturates prior to 10k steps and begins to oscillate afterward, so 10k serves as a stable and representative point for comparison rather than a selectively favorable one. Ideally, agents would be designed to continue improving without catastrophic forgetting, reducing the need for heuristic early-stopping decisions, a direction that aligns well with the broader goals of open-ended learning.
>
> We would further like to express our appreciation for the reviewer’s clarification questions, which prompted substantial improvements in the clarity and presentation of the manuscript. Your thoughtful comments were instrumental in refining the paper. We hope our responses satisfactorily resolve the concerns raised and kindly ask the reviewer to take them into consideration in their final evaluation.

---

### Official Review · Reviewer_S2xS · 2025-11-01

**Soundness:** 3
**Presentation:** 3
**Contribution:** 3
**Rating:** 6
**Confidence:** 3

**Summary:**

The paper proposes TRACED (transition-aware regret approximation with co-learnability for environment design), an unsupervised environment design framework based on not only positive value loss (PVL) but also average transition prediction loss (ATPL) and co-learnability. On MiniGrid and BipedalWalker, the method achieves strong performance with relatively few PPO updates.

**Strengths:**

By introducing an intuitive transition-prediction loss and co-learnability metric, the method yields strong empirical gains with inexpensive changes.

**Weaknesses:**

* While co-learnability is interesting and intuitive, the paper lacks theoretical analysis or guarantees.
* Despite its stochastic dynamics formulation, ATPL is computed based on deterministic predictions. For state–action pairs with high aleatoric stochasticity, the next state is inherently unpredictable, so ATPL can be large even when the future-value gap is negligible.

**Questions:**

Does training on tasks with high co-learnability at time $k$ continue to accelerate progress on other tasks at subsequent times? Could you quantify this temporal persistency of co-learnability? (e.g., reporting the covariance between $\mathrm{CoLearnability}_i(k)$ and $\mathrm{CoLearnability}_i(k+1)$)

---

> ### Author Response · Authors · 2025-11-24
>
> We thank the reviewer for the constructive and detailed assessment of our work. Our responses to each comment are provided below.
>
> ### Question
>
> > Q1. Does training on tasks with high co-learnability at time $k$ continue to accelerate progress on other tasks at subsequent times? Could you quantify this temporal persistency of co-learnability? (e.g., reporting the covariance between $\mathrm{CoLearnability}_i(k)$ and $\mathrm{CoLearnability}_i(k+1)$)
>
> Motivated by the reviewer’s question, we performed a dedicated temporal analysis on the multi-agent JaxNav benchmark (Appendix S). Our results show that Co-Learnability at time t exhibits a consistent positive correlation with Co-Learnability at time t ⁣+1, indicating that tasks with high Co-Learnability continue to provide transfer benefits at subsequent timesteps. This quantifies the temporal persistency of Co-Learnability.
>
> ### Additional Analyses
>
> In addition, Appendix R provides a theoretical analysis supporting the role of ATPL in regret approximation. We show that incorporating ATPL yields a tighter upper bound on the dynamics-induced regret mismatch, thereby providing a more faithful approximation to true regret. We also acknowledge that in environments with substantial aleatoric stochasticity, ATPL may exhibit higher variance. We have added this as a limitation and as an important direction for future work.
>
> We again thank the reviewer for the insightful feedback, which has allowed us to broaden our empirical analysis. We hope our responses satisfactorily resolve the concerns raised and kindly ask the reviewer to take them into consideration in their final evaluation.

---

### Official Review · Reviewer_3WBM · 2025-11-01

**Soundness:** 3
**Presentation:** 3
**Contribution:** 3
**Rating:** 4
**Confidence:** 4

**Summary:**

The authors propose TRACED, a novel curriculum selection criterion for Unsupervised Environment Design methods. TRACED consists of a novel, transition-aware regret approximation and adds a co-learnability term. They evaluate TRACED in standard UED benchmarks. They provide an ablation analysis over their proposed changes and additional analyses over the curricula differences between existing methods and TRACED. They claim to show improved performance over existing methods.

**Strengths:**

1. Overall, the writing is easy to follow and straightforward.
2. The transition-aware regret approximation is well motivated.
3. The notation is simple, and the definitions (Section 3.2) are easy to follow.
4. The figures are mainly well formatted and readable.
5. The authors provide fair computational comparisons and a wall-clock time comparison with most existing methods.

**Weaknesses:**

1. The authors do not rely on novel analysis tools as proposed by SFL [1], which would allow the authors to strengthen their claim that TRACED improves performance over existing methods. For example, the authors could have analysed CVaR to show robustness to worst-case levels. Similarly, SFL provides density maps, such as those in Figures 4 and 6 of the SFL paper, which would strengthen the claim of TRACED that they improve performance. It is also unclear to the reviewer why they do not compare their method with SFL. The authors state that SFL does not outperform PLR; however, this seems directly contradicted by the authors, as seen in Figure 7a and Figure 14b.
2. There appears to be some performance difference between PLR results reported at 10k in TRACED and SFT. In SFT, it appears that PLR can achieve a mean success rate of over 0.8 on the Minigrid evaluation set at 4k PPO update steps. In TRACED, the reported performance at 10k update steps is around 0.45. How can these performance differences be consolidated?


## Clarity
1. The authors could do a more thorough job of properly reporting statistically significant results and be more transparent in what they report. For example, Figure 7 present mean and standard deviation over clearly heavily skewed results, where standard deviation is not particularly meaningful, e.g., (12% +/- 28%). Again, median and IQR would have been more appropriate.

**Questions:**

1. Could you please clarify the error bars for each figure? For example, in Figure 5, what is the shaded area for the median or for the mean?
2. Could you please change the y-axis tick size in Figure 7?
3. Could you please clarify the performance differences between SFL's PLR and the author's implementation?
4. The reviewer typically tries to avoid suggesting extra experiments. However, given that MultiAgent JaxNav has been released for at least a year and it has been demonstrated that most existing methods perform worse than DR on JaxNav, I believe it's only fair to request an evaluation of TRACED on JaxNav against SFL and other existing methods. This is especially true if the authors claim to beat the "state-of-the-art" CENIE (line 67, 251, 475).
5. Could you clarify how the ATPL loss term is calculated? The reviewer is unsure if the LSTM that's used for the transition prediction is also the backbone model used to calculate the value function? Or is the LSTM transition prediction model a completely separate network and no weights are shared with he value function? The authors state in the appendix, line 1246, that they differ only in the LSTM design choice. Could they clarify what they mean?

To summarise, the paper is overall pretty solid, and the idea of including the trajectory loss to capture the distribution difference is definitely interesting. The paper may feel somewhat incremental, but that doesn't detract from its solid execution. While it's nice to know that the transition loss and co-learnability influence performance, it lacks a really in-depth explanation of why co-learnability or transition loss improves performance. The reviewer is looking forward to the rebuttal period and hearing back from the authors.

---

> ### Author Response · Authors · 2025-11-24
>
> We greatly appreciate the reviewer’s careful analysis and valuable suggestions, which have helped us further strengthen the paper. We respond to each of the raised points below.
>
> ### Questions
> > Q1. Could you please clarify the error bars for each figure? For example, in Figure 5, what is the shaded area for the median or for the mean?
>
> All shaded regions in the paper represent 95% confidence intervals. In Figure 7(c), the vertical error bars correspond to the standard deviation across seeds. These are now made explicit for clarity.
>
> > Q2. Could you please change the y-axis tick size in Figure 7?
>
> We have revised the presentation. In addition to the mean curves in the main paper, we now report four robust aggregate metrics, Median, IQR, IQM, and Optimality Gap, following the rliable protocol (Appendix P, Fig. 21).
>
> > Q3-4. Could you please clarify the performance differences between SFL's PLR and the author's implementation? The reviewer typically tries to avoid suggesting extra experiments. However, given that MultiAgent JaxNav has been released for at least a year and it has been demonstrated that most existing methods perform worse than DR on JaxNav, I believe it's only fair to request an evaluation of TRACED on JaxNav against SFL and other existing methods. This is especially true if the authors claim to beat the "state-of-the-art" CENIE (line 67, 251, 475).
>
> The performance discrepancy arises from the fact that the PLR implementation used in our work and the one reimplemented in SFL differ in multiple hyperparameters and architectural choices. For our experiments, we follow the ACCEL configuration (e.g., discount factor = 0.995, 5 PPO epochs, 256-dimensional hidden layers), whereas SFL employs an independently implemented JAX version of Robust PLR with substantially different settings (e.g., discount factor = 0.99, 4 PPO epochs, 512-dimensional hidden layers), among several other differences. These variations, across optimization settings, network capacity, and implementation details, jointly account for the observed discrepancies in PLR performance between the two works.
>
> To directly address the reviewer’s concern, we performed new experiments on the JaxNav-MultiAgent benchmark used in SFL, comparing TRACED, SFL, and Robust PLR under the SFL evaluation protocol (CVaR and mean win rate). Results are included in Appendix Q. TRACED consistently outperforms Robust PLR across all CVaR levels and achieves performance comparable to SFL. The full evaluation code for JaxNav-MultiAgent has been added to the supplementary materials.
>
> > Q5. Could you clarify how the ATPL loss term is calculated? The reviewer is unsure if the LSTM that's used for the transition prediction is also the backbone model used to calculate the value function? Or is the LSTM transition prediction model a completely separate network and no weights are shared with he value function? The authors state in the appendix, line 1246, that they differ only in the LSTM design choice. Could they clarify what they mean?
>
> The LSTM used for computing ATPL is a separate transition-prediction backbone and does not share parameters with the actor-critic network. Although parameter sharing is possible, it entangles the agent’s optimization objective with transition-prediction accuracy, creating potential gradient interference and making it difficult to isolate the effect of ATPL within the curriculum. To ensure a clean analysis of ATPL’s contribution, we intentionally use a distinct transition model.
>
> ### Additional Analyses
> In addition, we include in Appendix R a theoretical analysis demonstrating that incorporating ATPL into the regret approximation yields a tighter upper bound on regret mismatch, thereby providing a more faithful approximation to true regret. We also provide a detailed study of the temporal behavior of Co-Learnability, included in Appendix S, showing mild but consistent temporal persistence.
>
> We are grateful for the reviewer’s comments, which significantly improved both the theoretical discussion and experimental scope of the paper. We believe our clarifications address the raised questions, and we would appreciate the reviewer’s consideration of these responses in their overall judgement.

---

### Author Response · Authors · 2025-11-24

We sincerely thank all reviewers for their constructive and insightful feedback. In response to the raised concerns, we have made the following major improvements during the rebuttal period.

1. **Additional Experimental Evaluation**: We conducted new experiments on the JaxNav-MultiAgent benchmark used in SFL, comparing TRACED, SFL, and Robust PLR under the SFL evaluation protocol (CVaR and mean win rate). Results are included in Appendix Q.

2. **Theoretical Analysis of ATPL**: We included a new theoretical result showing that incorporating ATPL yields a tighter upper bound on regret mismatch, providing a more principled justification for its use in regret approximation. This is presented in Appendix R.

3. **Empirical Analysis of Co-Learnability**: We added a detailed study on the temporal consistency of Co-Learnability, quantifying its persistence across timesteps and reporting correlation statistics. This analysis is included in Appendix S.

We appreciate the reviewers’ thoughtful comments, which have helped us substantially strengthen both the theoretical and empirical contributions of the paper. We hope these improvements address the concerns raised and contribute to a clearer and more rigorous presentation.

---

### Meta-Review · Area_Chair_8LZa · 2025-12-22

**Summary:**

Reviewers raised concerns primarily regarding the lack of comparison with SFL and the absence of theoretical justification for the ATPL component. The authors addressed these concerns in the rebuttal by providing new experimental results on the JaxNav-MultiAgent benchmark used in SFL and by introducing a theoretical analysis showing that incorporating ATPL leads to a tighter upper bound on regret mismatch. These additions strengthen both the empirical and theoretical foundations of the paper. As a result, the major reviewer concerns have been satisfactorily addressed, supporting a positive recommendation.

**Reviewer Concerns:**

### Addressed
* **Additional experimental comparison with SFL**.
The authors conducted new experiments on the JaxNav-MultiAgent benchmark used in SFL, comparing TRACED, SFL, and Robust PLR under the SFL evaluation protocol. These results directly address reviewer requests for a fair and relevant comparison.

* **Theoretical analysis of ATPL**.
The authors included a new theoretical result showing that incorporating ATPL yields a tighter upper bound on regret mismatch, providing a more principled justification for the proposed method.

### Outstanding
* **None**. I do not identify any major outstanding concerns that would affect the overall contribution.

**Reviewer Scores:**

**Reviewer 3WBM (initial score: 4)**. This reviewer raised questions regarding clarification points and requested an additional comparison with SFL on JaxNav, while acknowledging the solid overall execution of the paper. The rebuttal clarifies these points and provides the requested experiments, in which TRACED continues to outperform the baselines. I would therefore expect the reviewer to increase their score after discussion.

**Reviewer S2xS (initial score: 6)**. This reviewer was generally positive but raised concerns about the lack of theoretical analysis and the quantification of temporal persistency of co-learnability. These issues are addressed in Appendix R and Appendix S of the revised version, respectively. I would expect the reviewer to maintain their positive assessment or slightly increase their score.

**Reviewer y3M6 (initial scores: 6)**. This reviewer was positive overall, raising clarification questions and requesting a comparison with SFL. The authors addressed the clarification points and added the requested experimental results in the rebuttal. I would expect the reviewer to maintain or increase their score.

**Reviewer mbPu (initial scores: 4)**. This reviewer responded prior to the ICLR data leak and explicitly stated that they would keep their current rating. I would therefore not expect a score change.

---

### Decision · Program_Chairs · 2026-01-26

Accept (Poster)